# Comparison of ozone profiles from DIAL, MLS, and chemical transport model simulations over Río Gallegos, Argentina during the spring Antarctic vortex breakup, 2009

Takafumi Sugita[1], Hideharu Akiyoshi[1], Elián Wolfram[2], Jacobo Salvador[2,4,5], Hirofumi Ohyama[3,1], and Akira Mizuno[3]

[1]National Institute for Environmental Studies, Tsukuba, Ibaraki, Japan
[2]Laser Application Research Center (CEILAP)-UNIDEF (MINDEF-CONICET), UMI-IFAECI-CNRS-3351, Villa Martelli, Argentina
[3]Institute for Space-Earth Environmental Research (ISEE), Nagoya University, Nagoya, Aichi, Japan
[4]Universidad Tecnológica Nacional, Facultad Regional Bs. As. (UTN-FRBA) Medrano 951 CABA, Argentina
[5]Universidad Nacional de la Patagonia Austral Unidad Académica Río Gallegos and CIT Santa Cruz, Argentina

*Correspondence to:* T. Sugita (tsugita@nies.go.jp)

**Abstract.** This study evaluates the agreement between ozone profiles derived from the ground-based DIfferential Absorption Lidar (DIAL), satellite-borne Aura Microwave Limb Sounder (MLS), and 3-D chemical transport model simulations (MIROC-CTM) over the South Patagonian Atmospheric Observatory (OAPA, 51.6°S, 69.3°W) in Río Gallegos, Argentina from September to November 2009. In this austral spring, measurements were performed in the vicinity of the polar vortex,

5  and inside it on some occasions; they revealed the variability in potential vorticity (PV) of measured air masses. Comparisons between DIAL and MLS were performed between 6 hPa and 100 hPa with 500 km and 24 h coincidence criteria. The results show a good agreement between DIAL and MLS with mean differences of $\pm 0.1$ ppmv (MLS – DIAL, n = 180) between 6 hPa and 56 hPa. MIROC-CTM also agrees to DIAL, with mean differences of $\pm 0.3$ ppmv (MIROC-CTM – DIAL, n = 23) between 10 hPa and 56 hPa. Both comparisons provide mean differences of 0.5 ppmv (MLS) to 0.8-0.9 ppmv (MIROC-CTM)

10  at the 83-100 hPa levels. DIAL tends to underestimate ozone values at this lower altitude region. Between 6 hPa and 8 hPa, the MIROC-CTM ozone value is 0.4-0.6 ppmv (5-8%) smaller than those from DIAL. Applying the scaled PV criterion for matching pairs in the DIAL/MLS comparison, the variability in the difference decreases 21-47% between 10 hPa and 56 hPa. However, the mean differences are slight for all pressure levels, except 6 hPa. Because ground measurement sites in the Southern Hemisphere are very sparse at mid- to high-latitudes, i.e., 35-60°S, the OAPA site is unique for evaluating the bias and

15  long-term stability of satellite instruments. The good performance of this DIAL system will be useful for such purposes in the future.

# 1 Introduction

Ozone depleting substances (ODS) have been decreasing since the Montreal Protocol and its subsequent adjustments and amendments. As a result, stratospheric ozone ($O_3$) is expected to increase in the future. The last WMO/UNEP ozone assessment concluded that increasing $O_3$ has been observed in the upper stratosphere around 42 km, or 2 hPa, in altitude (WMO, 2014).

Positive trends have been evaluated for both the tropics and 35-60° latitude bands of both hemispheres above 5 hPa levels from 2000 to 2016 (Steinbrecht et al., 2017). However, the trend is still not statistically significant below 10 hPa levels. Steinbrecht et al. (2017) found $0.7 \pm 0.9$ and $-0.2 \pm 1.4$% per decade changes at 10 hPa and 70 hPa, respectively, for 35-60°S. The satellite measurement has an advantage for estimating long-term trends because of its global coverage on a daily basis. However, its drift, i.e., the long-term measurement stability, should be quantitatively assessed with independent instruments. Ground-based

ozone lidar is a potential candidate for such purposes, and can be used to estimate drift (e.g., Nair et al., 2011, 2012; Eckert et al., 2014; Hubert et al., 2016).

Hubert et al. (2016) comprehensively evaluated the bias and drift of 14 limb-viewing satellite sensors using ozonesonde and ozone lidar measurements. They concluded that biases in the satellite sensors were within $\pm 5$% between 20 and 40 km and drifts were at most $\pm 5$% per decade. They suggested that several instruments have significant drifts; caution is needed for

multi-instrument comparisons to derive drift. Hubert et al. (2016) also showed a comparison spread, which is a measure of the short-term variability, with values of <5-12% for the same altitude range.

The ozone DIfferential Absorption Lidar (DIAL) system was installed at the South Patagonian Atmospheric Observatory (Observatorio Atmosférico de la Patagonia Austral, OAPA, 51.6°S, 69.3°W) in Río Gallegos, Argentina in 2005 (Wolfram et al., 2008). A map showing the OAPA site is shown in Figure 1. This site has been a stratospheric ozone lidar site within the

Network for the Detection of Atmospheric Compositions Change, NDACC (www.ndsc.ncep.noaa.gov) since December 2008. NDACC sites in the Southern Hemisphere (S.H.) are very sparse at mid- to high-latitudes (e.g., 35-60°S). In springtime, this site is occasionally inside the southern polar vortex, which has shifted off the pole or elongated. A long persistent coverage (~20 days) of the polar vortex over the southern tip of South America occurred in 2009 for the first time since 1979 (de Laat et al., 2010; Wolfram et al., 2012). In the 2009 austral spring between September and November, measurements at OAPA were

performed in the vicinity or, on some occasions, inside of the polar vortex, revealing a variability in potential vorticity (PV) of measured air masses inside and outside the vortex. Accordingly, the largest variability in $O_3$ values would be expected in such a latitude band (35-60°S). Therefore, this event was a good opportunity to assess the impact of $O_3$ variability on biasing behavior. To evaluate the performance of the DIAL system under such variability, $O_3$ data from Aura Microwave Limb Sounder (MLS) satellite measurements (Waters et al., 2006) were used for comparison. In addition to the DIAL/MLS comparison, we also used

$O_3$ values from a 3-D chemical transport model simulation, which is based on version 3.2 of the Model for Interdisciplinary Research on Climate (MIROC) (Akiyoshi et al., 2016). Therefore, a secondary objective of this study was to examine the performance of the model simulation. Measurement and model simulation data used here are described in Section 2. The methodology used for the comparison is provided in Section 3. Vertical profiles of $O_3$ and their time series at the two selected

pressure levels are shown in Section 4. The results of differences that depend on coincidence criteria are also shown in Section 4 and summarized for all pressure levels (from 6 hPa to 100 hPa). The conclusions of this study are described in Section 5.

## 2 Measurements and model simulations

### 2.1 Stratospheric ozone lidar

DIAL is a laser-based active remote sensing system operated from the ground, aircraft, and ship, and has a robust heritage (e.g., Mégie et al., 1977; Browell et al., 1983; Steinbrecht et al., 1989). $O_3$ measurements from DIAL have a high vertical resolution and measurements have shown long-term stability (Nair et al., 2012; Hubert et al., 2016), owing to the stratospheric ozone lidar sites of NDACC (e.g., Leblanc and McDermid, 2000; Brinksma et al., 2002; Godin-Beekmann et al., 2003; Steinbrecht et al., 2009). To target the stratosphere, $O_3$ number density is usually retrieved between 15 km and 45 km in geometric altitude. The

DIAL system installed at the OAPA site in Río Gallegos, Argentina, began operating in August 2005 (Wolfram et al., 2008). The DIAL system operated at this site is fully described in Wolfram et al. (2008) and included in a review (Antuña-Marrero et al., 2017). The instrument is briefly described here. The DIAL technique requires two emitted wavelengths generated by two emitter lasers. An excimer (XeCl) laser emitting at 308 nm with 30 Hz repetition rate and maximum energy per pulse of 300 mJ is used for ozone absorption. The reference wavelength corresponds to the third harmonic of the Nd-YAG laser

emission at 355 nm with 30 Hz repetition rate and maximum energy per pulse of 130 mJ. The optical receiver that collects the backscattered photons is composed of four Newtonian (f/2) telescopes defining an array of telescopes. Each has a 50-cm diameter with parabolic aluminized surfaces, 48-cm in diameter. This produces a total reception area of 7238 $cm^2$. An optical fiber, 0.27 db $km^{-1}$ with attenuation at 308 nm, is placed at the focus of each telescope. The other end of the fiber is positioned at the focus of a quartz lens placed inside a spectrometer used to separate the received wavelengths. A mechanical chopper is

positioned at the entrance of the spectrometer. It has a rotational frequency of 300 Hz (18,000 rpm), and its role is to block the strong lidar signals originating from the lower part of the atmosphere, typically below 10 km.

    The $O_3$ number density profile is computed using the DIAL equation from the difference between the signal slopes originating from Rayleigh scattering of the emitted laser beams ($n_{O3}$). Since the returned signals include scattering and attenuation by atmospheric molecules, aerosols, and other atmospheric components, this complementary term could be minimized with laser

wavelength chosen in the DIAL instrument. The laser wavelength chosen in the DIAL instrument minimizes the complementary term in the stratosphere to less than 10% of $n_{O3}$ measured, in the presence of low aerosol loading (Pelon et al., 1986). Because lidar signals cover a large dynamic range, they have to be attenuated for measurements in the lower stratosphere. Therefore, the final $O_3$ profile corresponds to a composite profile computed from the "low" and "high" energy Rayleigh signals which are detected simultaneously (e.g., Godin et al., 1989).

In the 2009 spring, measurements began on September 6 (UTC, Coordinated Universal Time) during clear-sky local night-time. Because the latitude of OAPA is 51.6°S, the short night lengths with increased seasonal cloud cover made it challenging to perform measurements after December (Wolfram et al., 2012). In total, 23 vertical profiles of ozone were obtained between September and November 2009, which were used for this study. Most measurements were performed for 3-5 h to obtain a

good signal-to-noise ratio (see Table S1 for detailed numbers). If we assume some typical wind speed of 30 m/s wind speed in the lower stratosphere, a horizontal spatial resolution becomes 300-500 km. In actual, we have evaluated horizontal distances using air-parcel trajectory analysis at 83 hPa (Tomikawa and Sato, 2005) and the results are summarized in Table S1. The actual vertical resolution ranged from 0.7 km to 4 km at 14 km and 35 km in altitude, respectively. The total measurement uncertainty also ranged from 3% to 15% at the same altitudes.

For the total measurement uncertainty (Wolfram et al., 2008), we evaluated the effect of ozone absorption cross section, which is temperature dependent, and found the error is not larger than 2%. The other source is from correction of aerosol contamination. The methodology uses a Fernald inversion algorithm to evaluate the aerosol backscatter signal at 355 nm and extrapolated to 308 nm. In order to increase the signal to noise ratio, the signal registered is averaged over the full acquisition time of the measurement. The acquisition time is typically three to four hours, according to weather conditions. Before processing the signal using the DIAL equation, we make two corrections: 1) subtraction of the background signal using a linear regression within the range of altitudes where the lidar signal is considered negligible, typically between 80 and 150 km; 2) dead time correction of the detector, in order to correct the saturation of the photocounting signals (pile-up effect) in the lower altitude ranges (Godin et al., 1999).

## 2.2 Aura MLS

The MLS measurement covers latitudes between 82°N and 82°S since August 2004 (Waters et al., 2006). It is onboard the National Aeronautics and Space Administration (NASA) Earth Observing System (EOS) Aura satellite. MLS measures millimeter- and submillimeter-wavelength thermal emission from the limb of the Earth's atmosphere every 25 seconds, from which vertical profiles of more than 15 chemical species are retrieved. We used the standard $O_3$ data product (240 GHz radiances) retrieved with the version 4.2 data processing algorithm, which is publicly available from mls.jpl.nasa.gov/. The quality of the $O_3$ data is as follows from Livesey et al. (2017). The vertical × horizontal resolutions are 3 km × 300 km at 100 hPa and 3 km × 500 km at 10 hPa. The precision is 0.03 ppmv at 100 hPa and 0.1 ppmv at 10 hPa. The accuracy estimated from systematic uncertainty characterization tests are 0.05 ppmv at 100 hPa and 0.3 ppmv at 10 hPa. Data screening was accomplished according to Livesey et al. (2017). The former versions of the MLS $O_3$ values are evaluated from comparisons with DIAL (Jiang et al., 2007). They showed a good agreement of ∼5% from 5 hPa to 100 hPa.

## 2.3 Nudged chemistry-climate model based on MIROC3.2 GCM

As described in Akiyoshi et al. (2016), NIES developed nudged chemistry-climate models (CCMs) using the MIROC model. The CCM was nudged toward European Center for Medium-Range Weather Forecasts (ERA)-Interim data below 1 hPa (Dee et al., 2011). In these nudged-CCMs, a set of model variables for zonal-wind ($u$), meridional wind velocities ($v$), and temperature ($T$) were nudged. Above 1 hPa, where no ERA-Interim pressure level data exist, the zonal-means of zonal wind and temperature are nudged toward the COSPAR international Reference Atmosphere 1986 data (CIRA, 1990). The time scale for nudging the meteorological data ($u$, $v$, and $T$) was set to one day.

The model used in this study is a spectral model with T42 horizontal resolution ($2.8° \times 2.8°$) and 34 vertical atmospheric layers above the surface. The top layer is located at approximately 80 km (0.01 hPa). Hybrid sigma-pressure coordinates are used for the vertical direction. The chemical constituents included in this model are $O_x$, $HO_x$, $NO_x$, $ClO_x$, $BrO_x$, hydrocarbons for methane oxidation, heterogeneous reactions for sulfuric-acid aerosols, supercooled ternary solutions, nitric-acid trihydrate, and ice particles. The CCM contains 61 chemical constituents including 42 for prediction and 19 for photochemical equilibrium, 165 gas-phase reactions, 42 photolytic processes, and 13 heterogeneous reactions on multiple aerosol types. The reaction rates and absorption coefficients are based on JPL 15-10 (Burkholder et al., 2015). The bromine budget is increased for consistency with observations using additions of $CHBr_3$ and $CH_2Br_2$, which results in approximately 21 pptv total inorganic bromine, $Br_y$, in the stratosphere around the year 2000. The volume mixing ratio of total inorganic chlorine, $Cl_y$, is approximately 3.3 ppbv in the stratosphere over the same period. This nudged CCM is hereafter termed the MIROC-chemical transport model (MIROC-CTM).

## 3 Method for comparisons between DIAL and MLS/CTM

The $O_3$ profiles from DIAL are used to evaluate the bias and drift, i.e., long-term stability of satellite measurements (e.g., Nair et al., 2012; Eckert et al., 2014; Hubert et al., 2016). Therefore, it is important to show the quality of the respective ground-based DIAL performance. Although $O_3$ profiles at OAPA are included in Hubert et al. (2016), the result for OAPA alone is not shown. Wolfram et al. (2012) also did not show coincident $O_3$ profiles with any limb-viewing satellite instruments. Therefore, we revisit the quality of the DIAL $O_3$ profiles obtained in the 2009 austral spring.

Usually, comparisons between DIAL and limb-viewing satellite instruments are conducted considering the differences in their vertical resolution and retrieval strategies (Hubert et al., 2016). MLS has covered the location of OAPA (51.6°S) on a daily basis since measurements began in 2005. The long-term stability of the MLS ozone dataset has been shown to be very good (e.g., Nair et al., 2012; Hubert et al., 2016). For comparison between DIAL and MLS, the DIAL profile is convolved using the following equation (Livesey et al., 2017):

$$X_{comv} = X_a + A[X_{DIAL} - X_a] \tag{1}$$

where $X_a$ is the a priori profile for each retrieval and $A$ is the averaging kernel functions (matrix) of MLS. $X_{DIAL}$ is the DIAL ozone profile, and $X_{comv}$ is the convolved DIAL ozone profile, which is converted to each MLS grid for comparison. We used $A$ for the polar winter condition from two $A$ that have been provided in the MLS dataset, the other is for the tropical condition.

We used 500 km in distance, the great circle (between 47.1°S and 56.1°S for 69.3°W) , and ±24 hours for coincidence criteria between DIAL and MLS measurements. Because the mid-point for the DIAL measurement duration was usually 2-3 UTC, the time differences (MLS – DIAL) were 0-4 hours or 13-17 hours on the same day that correspond to night or day paths of the EOS-Aura orbit. When no MLS measurements were available on the same day (9 cases), measurements one day before were used. In those cases, the time differences were –6 to –10 h or –20 to –24 h. For the DIAL measurement on October 27, an MLS measurement on October 28 was used, resulting in a 26 h difference. Both DIAL measurements on October 7 and 8 used ten MLS measurements on October 7 for matching pairs. In total, 180 matching pairs were used in this study.

For comparisons between DIAL and MIROC-CTM, we also unified the vertical grids for comparison. The DIAL profiles were linearly interpolated onto the pressure grids for the MLS data; the vertical increments of the DIAL profile are as small as 150 m. The MIROC-CTM profiles on the day of each DIAL measurement were interpolated onto the pressure grids for the MLS data using a cubic-spline. Both interpolated values were used to compute differences (MIROC-CTM – DIAL) (see Figures 2, 5, and 6).

For converting the original DIAL geometric altitude and $O_3$ number density to pressure and $O_3$ mixing ratio, the NCEP reanalysis data (Kalnay et al., 1996) are used. These data are registered in the NDACC database. Possible deviations could be expected if we use other meteorological data for the conversion process in DIAL. However, in this study, we used the DIAL data that registered in the NDACC database. Another possible deviations could also be expected if we use other meteorological data for the nudging process in MIROC-CTM. The different reanalysis data may cause different vertical and horizontal motions of air in the model, providing different tracer correlations, hence ozone field. However, in this study, we analyze owing to the model of Akiyoshi et al. (2016) to examine the performance.

## 4 Results and discussion

### 4.1 Example of vertical profile comparison

Figure 2a shows vertical profiles of $O_3$ measured with DIAL compared with those of MLS on the same day (November 14, 2009) as an example. The plus-crosses and dotted-line show the converted DIAL profile using Equation (1) and the original high vertical resolution DIAL profile, respectively. Each MLS profile was color-scaled with its measurement latitude to observe the latitudinal difference between DIAL and MLS. The bar in MLS $O_3$ profiles shows the precision reported for individual profiles. The bar in the DIAL $O_3$ profile shows the total uncertainty. The combined uncertainty (root sum square) is shown in the right panel. In addition to the DIAL and MLS profiles, we also compared the 24 h average $O_3$ profiles from MIROC-CTM at 12 UTC. We have extracted data from six locations between 48.8°S and 54.4°S in latitude at 67.5°W and 70.3°W in longitude, but the nearest grid data was plotted in Figure 2a (see Figures 5 and 6 for the variability in six model grids).

On this day, the DIAL profile above 50 hPa, i.e., pressures smaller than that level, revealed lower $O_3$ values, which was suggested in Wolfram et al. (2012) due to the edge of the southern polar vortex located near OAPA on November 14. Wolfram et al. (2012) also suggested that an altitude region around a potential temperature (PT) of 650 K was just inside the vortex. Several PT levels corresponding to pressure are also shown as text in Figure 2a. This DIAL profile agrees well with MLS profiles observed at similar latitudes, 51.7°S with green lines. The MLS profiles revealed a larger latitudinal difference of ~2.5 ppmv over ~8° especially at ~50 hPa level. For reference, the MIROC-CTM profiles also revealed latitudinal differences of ~1 ppmv over 5.6° at the same pressure level (not shown), suggesting a weaker latitudinal gradient in the model simulation for these conditions. In addition, the MIROC-CTM $O_3$ value is higher than from DIAL around 20 hPa levels. We will discuss this feature in the MIROC-CTM in Section 4.2.

In the right panel of Figure 2a, the differences between MLS $O_3$ and DIAL (MLS – DIAL) are shown. In addition, the difference between DIAL and the nearest MIROC-CTM is shown. In general, the MLS profiles of similar latitudes (51.7°S)

with OAPA are in good agreement with the DIAL profile within $\pm 0.5$ ppmv between 100 hPa and 6 hPa. The largest negative value is found at 46 hPa, with 2.0 ppmv for a profile of the highest latitude measured (54.7°S). In contrast, the largest positive value is found at 22 hPa, with 1.2 ppmv for a profile of the lowest latitude measured (48.8°S). This indicates that lower $O_3$ values still exist inside the vortex, i.e., depleted ozone in the spring time has not yet recovered, and larger $O_3$ values are found

outside the vortex at the lower latitudes in the middle stratosphere.

Another example is shown in Figure 2b. On this day, November 23, there were less latitudinal differences in ozone field compared to the result on November 14 as observed by MLS. Consequently, the latitudinal difference in MIROC-CTM is also smaller on November 23 than on November 14 (not shown). Similar to the former result, the MLS profile at a similar latitude with OAPA is in good agreement with the DIAL profile within $\pm 0.5$ ppmv between 83 hPa and 6 hPa. Whereas, the

MIROC-CTM $O_3$ is lower than DIAL by $\sim$–2 ppmv between 10 hPa and 6 hPa. This is discussed in Section 4.4.

## 4.2 Time series comparison

All 23 DIAL profiles obtained in September-November 2009 were evaluated for their variability with time. The PV values at the location and time of all $O_3$ profiles from DIAL, MLS, and MIROC-CTM were investigated to place the measurements inside or outside the polar vortex. The degree of PV values at each measurement or model grid is a robust indicator of the location

relative to the polar vortex. Here, we used meteorological data from the NASA Global Modeling and Assimilation Office (GMAO) Modern-Era Retrospective Analysis for Research and Applications-2 (MERRA-2) reanalysis (Molod et al., 2015; Gelaro et al., 2017) (gmao.gsfc.nasa.gov/reanalysis/MERRA-2/). We calculated the scaled PV (sPV) for pressures between 100 hPa and 6 hPa from the PV values from MERRA-2 and PV/sPV ratios as a function of PT. The PV and sPV values are provided through the MLS website as the derived meteorological products (DMPs) (Manney et al., 2007). We used version 2 of

DMP (GEOS5MERRA2 for the version 4 MLS data). sPV values ($s^{-1}$) are nearly constant at levels throughout the stratosphere (e.g., Dunkerton and Delisi, 1986; Manney et al., 1994, 2007). Figure 3 shows all the 23 profiles of $O_3$ obtained by DIAL. Data are color-scaled based on sPV values. The difference in the $O_3$ value is found depending on the sPV value especially above 30-40 hPa. Figure 4 shows sPV maps from MERRA-2 for selected days on September 26, October 3, November 14, and November 23, 2009. At 20 hPa, the polar vortex significantly diminishes on November 23 compared to that on September 26.

Whereas at 50 hPa, the polar vortex still exists on November 23 with smaller area than that on September 26.

An sPV value of $\sim 1.4 \times 10^{-4}$ $s^{-1}$ has been used to define the Northern Hemisphere (N.H.) polar vortex edge center (e.g., Ryan et al., 2016, and references therein). In addition, values of $\sim 1.6$ and $\sim 1.2 \times 10^{-4}$ $s^{-1}$ have been used to define the inner and outer edges, respectively. Those vortex edge definitions, i.e., center, inner, and outer, are according to Nash et al. (1996). We examined these values using the DMPs for the MLS measurements for the period studied here (i.e., September to November

2009). The results were somewhat different from those from the N.H. depending on time and altitude. For example, center, inner, and outer boundaries are defined by the absolute sPV values of 1.6, 1.9, and $1.3 \times 10^{-4}$ $s^{-1}$ at 68 hPa in November. The sPV values shown in the following figures are useful guides for showing positions relative to the vortex.

As representatives for the middle and the lower stratospheres, results at 18 hPa and 56 hPa are shown in Figure 5 and Figure 6, respectively. Figure 5a shows the time variation of $O_3$ values obtained from DIAL and MLS at 18 hPa. Both $O_3$

values are color-scaled using sPV values. On several occasions, $O_3$ values below 4 ppmv were measured by DIAL in air masses with larger sPV values, i.e., larger negative values indicated with blue and purple colors, in conjunction with the polar vortex dynamics.

For both September 26 and October 5, the polar vortex shifted toward the South American side, covering the OAPA site. On November 13-14, the $O_3$ values were low again. Correspondingly, the MLS $O_3$ values also show lower values with higher sPV values. In general, the DIAL $O_3$ values are within the variations of MLS $O_3$ values for each coincident date during all comparison periods. To quantitatively evaluate the degree of agreement, the differences between the two (MLS – DIAL) are shown in Figure 5c. These values are color-scaled using the sPV value from each MLS measurement. We computed mean and root-mean-square (rms) differences of $O_3$ from all 180 data points. At 18 hPa, the mean difference is –0.03 ppmv and the rms difference is 0.78 ppmv. Although the mean value shows a good agreement, the variance is large especially in September. We will discuss this large variance in Section 4.3.

Figure 5b for 18 hPa also shows time variations in $O_3$ values obtained from DIAL and those simulated with MIROC-CTM. Figure 5d shows the $O_3$ differences between DIAL and MIROC-CTM (MIROC-CTM – DIAL). In this plot, mean and rms differences in $O_3$ are calculated from all data points of the nearest model grid, 51.6°S/70.3°W, to the OAPA site (the number is 23). As a result, the mean difference is 0.04 ppmv and rms difference is 0.72 ppmv. For reference, Figures 5e and 5f show the relative differences for DIAL/MLS and DIAL/MIROC-CTM comparisons, respectively.

Similar to the DIAL-MLS comparison, both the DIAL and MIROC-CTM $O_3$ values show low values with larger sPV values, which indicate that the locations are inside the polar vortex, or that the air masses originate from the polar vortex. However, MIROC-CTM overestimates $O_3$ values with the larger sPV values compared to DIAL. When those higher deviations in MIROC-CTM are found, the DIAL $O_3$ values show smaller amounts below $\sim$4 ppmv (Figure 5b). This is also observed in the vertical profile in Figure 2a. The overestimate of MIROC-CTM may be partly due to the relatively coarse horizontal resolution of the model with regard to a complicated spatial structure near the boundary of the polar vortex in the breakup season. The polar vortex begins to breakup at higher altitudes, and then propagates downward. Another possible explanation could be due to a weaker vertical motion of air in MIROC-CTM. Although not shown, a vertical profile of nitrous oxide, $N_2O$, from MIROC-CTM on November 14, 2009 is different from that from MLS. A tight correlation between $N_2O$ and $Cl_y$ is found in the stratosphere (e.g., Schauffler et al., 2003), and used to infer the $Cl_y$ value from a measured $N_2O$ value (e.g., Wetzel et al., 2010; Strahan et al., 2014). At 18 hPa, the MIROC-CTM $N_2O$ value is higher than that of MLS, resulting in a smaller value of $Cl_y$ in MIROC-CTM. Thus, a smaller active chlorine ($ClO_x$) induces a higher $O_3$ amount in MIROC-CTM.

Figures 6a and 6c show time variations in $O_3$ values from DIAL and MLS, and the difference between the two at 56 hPa, similar to Figures 5a and 5c. Figures 6b and 6d also show time variations in $O_3$ values at 56 hPa from DIAL and MIROC-CTM, and the difference between the two, similar to Figures 5b and 5d. Figures 6e and 6f show the relative differences for DIAL/MLS and DIAL/MIROC-CTM comparisons, respectively. Unlike the characteristics of the 18 hPa result, significant lower ozone values relative to the other dates were not found inside the polar vortex on September 26 and October 5. Whereas, on November 13-14 and 23-24, lower $O_3$ values inside the polar vortex were found from both of DIAL and MLS. This is in agreement with the long-lasting polar vortex dynamics in the 2009 spring (de Laat et al., 2010; Wolfram et al., 2012). The mean

differences between DIAL and MLS/MIROC-CTM are as small as 0.06 ppmv and 0.16 ppmv, respectively. The rms differences are 0.46 ppmv and 0.36 ppmv for DIAL/MLS and DIAL/MIROC-CTM comparisons, respectively, which are smaller values than those at 18 hPa. The overestimate of MIROC-CTM with larger sPV values, as seen at 18 hPa is not evident at 56 hPa. One explanation may be that the polar vortex is more stable at 56 hPa than at 18 hPa, even on November 23-24.

## 4.3 Dependency in distance and sPV difference

The good correlation between sPV and $O_3$ values near the vortex boundary in austral spring has been previously shown in satellite measurements (e.g., Manney et al., 1999, 2001, 2005). Therefore, a horizontal gradient in $O_3$ should have been present at the vortex boundary in the 2009 spring. A previous study suggested that a better agreement is found when the comparison is performed with matching meteorological conditions using parameters such as sPV and equivalent latitude (Manney et al., 2001). Therefore, we further examined the larger variability between DIAL and MLS at 18 hPa, from the perspective of different sPV values. Figure 7a shows the $O_3$ difference (MLS – DIAL) versus sPV difference between DIAL and MLS (MLS – DIAL). Similar to Figure 5b, the data points are color-scaled based on the sPV values of the MLS measurements. A positive correlation between $O_3$ and sPV differences is found, suggesting lower $O_3$ values in MLS (negative in the y-axis) with a more poleward MLS profile, i.e., negative in the x-axis. Conversely, higher $O_3$ values in MLS, i.e., positive in the y-axis, with the lower latitude side profile in MLS, i.e., positive in the x-axis, is also seen, although the correlation is weaker than in the negative value area. After filtering out matching pairs over a certain sPV difference, e.g., below or above $\pm 0.3 \times 10^{-4}$ s$^{-1}$, the rms difference between DIAL and MLS at this pressure level decreases significantly. Such an sPV criterion is useful for suppressing the large rms difference found in $O_3$ measurements affected by the motion of polar vortex. Whereas, the mean difference less changes applying such the sPV criterion. This is consistent with the result from Holl et al. (2016) who showed differences in $CH_4$ values observed in the northern high latitude and sPV criterion with a value of $0.2 \times 10^{-4}$ s$^{-1}$ has little effect below 25 km in altitude.

We also examined results from 56 hPa in Figure 7b. Similar to the results from 18 hPa, larger $O_3$ differences are found with larger sPV differences. Applying certain sPV criterion to these data, the mean difference changes only slightly, but the rms difference decreases, similar to the results from 18 hPa. The results for other pressure levels are summarized in Section 4.4.

Since the MERRA-2 data set also provide the $O_3$ value (Wargan et al., 2017), we examined those data instead of the sPV value. Figure 8 shows the $O_3$ difference versus MERRA-2 $O_3$ difference between DIAL and MLS (MLS – DIAL). The mean difference is computed from the horizontal axis, resulting in –0.12 ppmv at 18 hPa and –0.02 ppmv at 56 hPa. The measured $O_3$ difference is well reproduced by the MERRA-2 $O_3$ that assimilates Aura MLS as well. At 56 hPa, a compact correlation is found between the two differences with a slope of one-by-one. A similar positive correlation is also found at 18 hPa.

In addition to the sPV differences examined, we evaluated the correlation between the $O_3$ difference and distance in the DIAL/MLS measurements (Figure 9). In these figures (Figure 9a for 18 hPa and Figure 9b for 56 hPa), data points are color-scaled based on the sPV difference between DIAL and MLS (MLS – DIAL). Clearly, larger $O_3$ differences, especially those with negative values in Figure 9a, have large sPV differences, i.e., below $-0.5 \times 10^{-4}$ s$^{-1}$). As shown in the figures, the $O_3$ difference does not depend critically on the distance between the two measurements.

In summary, the O$_3$ differences between DIAL and MLS can be partly attributed to differences in the measurement points. Furthermore, the O$_3$ difference is more correlated with sPV differences between than the difference in distance. Therefore, it is important to analyze O$_3$ values with sPV (or PV) values near the polar vortex boundary, which has been suggested previously (e.g., Manney et al., 2001).

## 4.4 Comparison at other levels: summary

The mean and rms differences computed from the time-series comparisons in Section 4.2 were extended for other pressure levels to summarize the degree of agreement between DIAL and MLS or MIROC-CTM. These results are plotted versus pressure in Figure 10. Absolute differences are shown in the left panel. Relative differences, the absolute differences divided by their mean values of O$_3$, are shown in the right panel. In the left panel, mean differences (open circle and cross) for both DIAL/MLS and DIAL/MIROC-CTM comparisons, along with rms differences (dotted lines) are shown. The mean differences of the DIAL/MLS comparison are almost within ±0.1 ppmv between 6 hPa and 56 hPa with 180 data points for each level. This corresponds to the relative values, in the right panel, of ±3%. Figure 11 shows differences between DIAL and MLS using the sPV criterion. The mean and rms differences shown in this figure as blue lines are identical to Figure 10. The mean and rms differences after filtering with the sPV criteria ($\pm 0.3 \times 10^{-4}$ s$^{-1}$) are shown as green lines. Clearly, the rms differences decrease 21-47% between 10 hPa and 56 hPa; the number of data points was reduced from 146-180 to 107-144. However, the mean differences only change slightly for all pressure levels, except for the 6 hPa level.

For the DIAL/MIROC-CTM comparison, the mean differences are almost within ±0.3 ppmv between 10 hPa and 56 hPa, with 23 data points for each level. This corresponds to relative values of ±8%. Above 8 hPa, the absolute differences increase to –0.6 ppmv which corresponds to relative values of –8%. To examine the low bias in MIROC-CTM, the time-series in O$_3$ difference between DIAL and MIROC-CTM at 8 hPa is shown in Figure 12. Larger negative deviations in MIROC-CTM are found in October and November, especially for data with sPV values between –1.0 and $-1.5 \times 10^{-4}$ s$^{-1}$. Similar results are also found from 6 hPa and 7 hPa levels. The peak altitude of ozone in MIROC-CTM is lower than that of DIAL, as shown in Figure 2. Both the vertical and horizontal motions of air in the model are responsible for this different feature, but the cause is not known. As was shown in Figure 3, the vertical gradient of O$_3$ from DIAL above 15-20 hPa shows rather week inside the polar vortex, but occasionally strong outside or edge of the polar vortex. Thus, the vertical gradient of O$_3$ may affect the result for such occasions with the steeper gradient. The feature presented here suggests a difficulty in the reproduced ozone field for those pressure levels (6-8 hPa) in these latitudes and season using this version of MIROC-CTM. As discussed in Section 4.2, the polar vortex breakup process may cause a highly variable spatial structure. This may be partly responsible for the difference because of the insufficient spatial resolution of the model to distinguish this process.

Both the DIAL/MLS and DIAL/MIROC-CTM comparisons show increasing rms differences with increasing altitudes above the 20-30 hPa levels, reaching more than 1 ppmv. This is partly due to the O$_3$ value increasing with increasing altitudes. Thus, relative values of the rms difference (Figure 10b) do not show strong vertical gradients compared to the absolute values (Figure 10a).

Both comparisons also show larger absolute differences below 68 hPa, reaching 0.5 ppmv (116%) for DIAL/MLS and 0.9 ppmv (292%) for DIAL/MIROC-CTM. This suggests a lower bias in the DIAL measurement at these lower altitudes (∼80-100 hPa) of some magnitude. As discussed in Wolfram et al. (2008), this DIAL system has some difficulty in measuring around 100 hPa and below due to saturation from backscattered photons in the low-energy channels. Since the $O_3$ mixing ratio from DIAL is very small below about 70 hPa, the sensitivity might be degraded along with the saturation effect. Therefore, DIAL data at this altitude range should be used with caution.

Another possible reason is the difference in measured ozone associated with the difference in original vertical resolution, ∼1 km for DIAL versus 3 km for MLS. In this period, lamina structures in $O_3$ profiles are often observed from ozonesonde measurements, especially below 20 km. DIAL may capture lower values of $O_3$ in these lamina structures while collecting measurements over 3-5 h, compared to MLS that measures instantaneously along the orbit, nearly the north-south direction (see Supplement). This may facilitate $O_3$ differences, to a certain extent, even while both measurements are accurate. In the other geophysical regions of the Asian monsoon anticyclone, difficulties in MLS retrievals within the strong vertical gradient of $O_3$ have been discussed (Yan et al., 2016). The largest $O_3$ difference between DIAL and MLS at 83 hPa was found on October 3, 2009; this case was studied using air mass trajectory analysis (Tomikawa and Sato, 2005) and the $O_3$ field from MIROC-CTM (see Supplement).

## 5   Conclusions

Ground-based DIAL measurements were performed at the OAPA in Río Gallegos (51.6°S, 69.3°W), Argentina, from September to November 2009, when a long-lasting southern polar vortex, and accompanying ozone depletion, occurred over the area for the first time since 1979 (de Laat et al., 2010; Wolfram et al., 2012). This site is one of the few NDACC DIAL sites in the S.H. Focusing on this period of large dynamical variability in measured air masses during the movement of the polar vortex, it is possible to analyze the effects of the polar vortex on $O_3$ variability. Twenty-three $O_3$ profiles were obtained by DIAL during the period. These profiles were compared with coincident MLS $O_3$ profiles with 180 matching pairs, based on time and space criteria.

The mean differences between DIAL and MLS are within ±0.1 ppmv (±3%) from 6 hPa to 56 hPa, showing good agreement regardless of the large sPV variability between each matching pair. The DIAL data are also compared with outputs from the MIROC-CTM model simulation. The mean differences between DIAL and MIROC-CTM are within ±0.3 ppmv (±8%) from 10 hPa to 56 hPa. Above 8 hPa, the mean differences increases to –0.6 ppmv (–8%). To measure variability in the comparison, rms differences between DIAL and MLS or MIROC-CTM are also evaluated. For both DIAL/MLS and DIAL/MIROC-CTM comparisons, the rms differences are nearly 0.5 ppmv for pressure levels between 30 hPa and 100 hPa, and increase with increasing altitudes up to 6 hPa, reaching 1.1-1.2 ppmv. From the DIAL/MLS comparison, the $O_3$ differences depend on sPV differences at 18 hPa. Therefore, another criterion for comparison is proposed: pairs with absolute sPV differences that exceed $0.3 \times 10^{-4}\,\mathrm{s}^{-1}$ are discarded. As a result, the rms differences decreased significantly between 10 hPa and 56 hPa, but the mean differences only slightly change for all pressure levels, except for 6 hPa.

The comparison between DIAL and MLS indicates that the $O_3$ difference is partly due to sPV differences between measurement locations; however as yet unknown factors create additional differences. The comparison between DIAL and MIROC-CTM indicates that an insufficient model spatial resolution may be partly responsible for the $O_3$ differences above 18 hPa during polar vortex breakup. An insufficient model vertical motion may also be partly responsible for the $O_3$ differences, especially inside the polar vortex. Both the DIAL/MLS and DIAL/MIROC-CTM comparisons also show larger mean differences below 68 hPa, reaching 0.5 ppmv (116%) and 0.9 ppmv (292%) at 100 hPa, respectively. One possible cause may be a low bias in the DIAL $O_3$ measurement, but this hypothesis was not confirmed in this study. Nevertheless, finding good agreement between DIAL and MLS $O_3$ measurements between 6 hPa and 56 hPa is a necessary step for studies in evaluating bias and long-term stability of satellite sensors in the future. Because of very sparse observations from S.H. ground-based stations, continuation for long-term measurements there for NDACC is highly recommended. This study provides an outlook for continuing measurements at the OAPA site. The DIAL measurements at the OAPA site are available for all years since 2005, except 2016 when no measurements were collected. The result of the DIAL/MLS comparison using these long-term data will be published elsewhere.

*Acknowledgements.* This research was supported by the Science and Technology Research Partnership for Sustainable Development (SATREPS), Japan Science and Technology Agency (JST) and Japan International Cooperation Agency (JICA). The DIAL construction and maintenance from 2005 until the present were supported by projects lead by Eduardo Quel (CEILAP), type PICT from the Science and Technology Ministry of Argentine, and in collaboration with CONICET, and the late Gérard Mégie and Sophie Godin-Beekmann (CNRS, France), and Hideaki Nakane (Kochi University of Technology, Japan). The Aura MLS data were produced at the Jet Propulsion Laboratory (JPL), California Institute of Technology, under contract with the National Aeronautics and Space Administration (NASA). CTM computations were conducted on NEC-SX9/A(ECO) computers at CGER, NIES. We thank Yousuke Yamashita (Japan Agency for Marine-Earth Science and Technology), and Haruna Nakamura and Izumi Ikeuchi (Fujitsu FIP Corp., Japan) for configuring and preparing the model computations. The derived meteorological products (DMPs) were produced by Gloria L. Manney (NorthWest Research Associates and New Mexico Institute of Mining and Technology) and Luis F. Millán (NASA/JPL). We thank Masanao Kadowaki (Japan Atomic Energy Agency) for his helpful discussion in the early stages of this manuscript. The air mass trajectories were computed by the National Institute of Polar Research, Japan, trajectory model using the NASA GMAO's MERRA data. We would like to thank Editage (www.editage.jp) for English language editing.

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

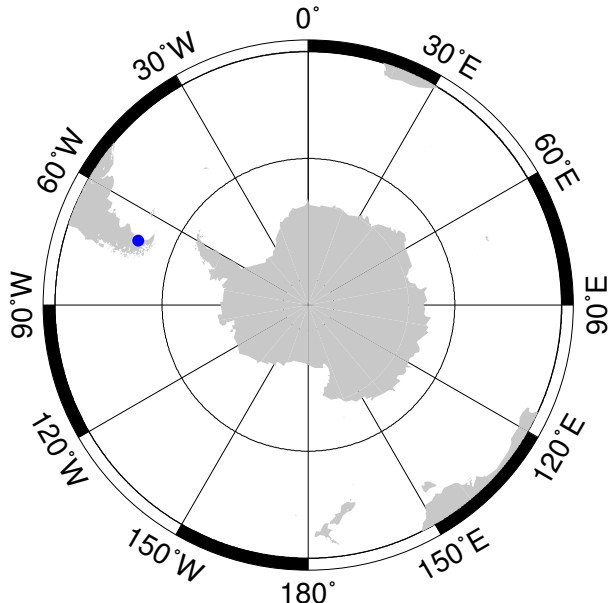

**Figure 1.** Location of the OAPA site in Río Gallegos, Argentina (51.6°S, 69.3°W), shown as a blue circle. Latitude ranges from 30°S to 90°S.

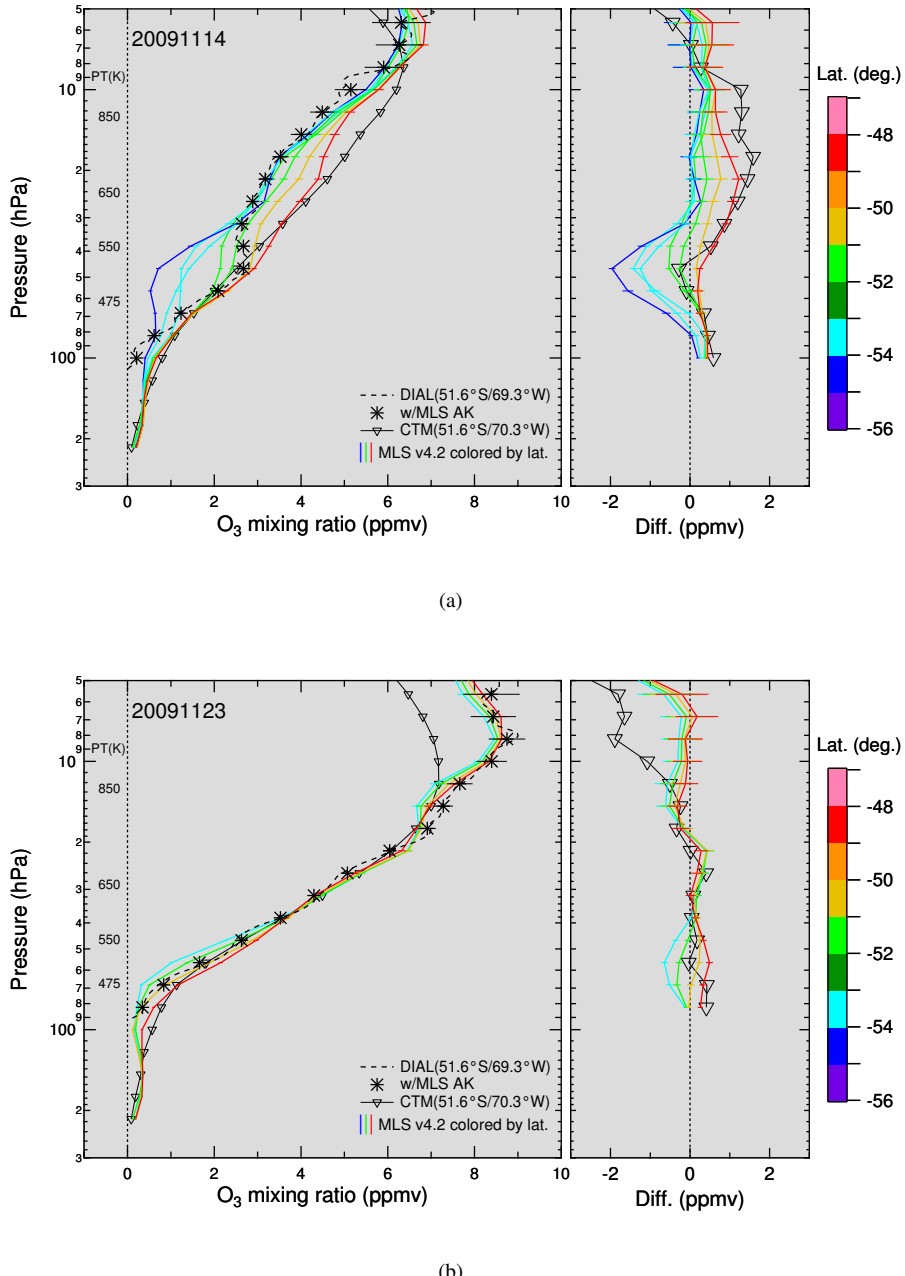

**Figure 2.** Vertical profiles of O$_3$ mixing ratios on November 14, 2009 (a) and November 23, 2009 (b) measured using DIAL (plus-crosses and dotted-line) and MLS (solid-lines with color) over the OAPA site (see text for additional description). A MIROC-CTM O$_3$ profile of nearest grid for the OAPA site is also shown. Corresponding potential temperatures for pressure are shown as text in the vertical axis. Differences between DIAL and X (MLS or MIROC-CTM) (X – DIAL) are shown in the right panel (see text). The MLS profiles are color-scaled based on their measurement latitudes.

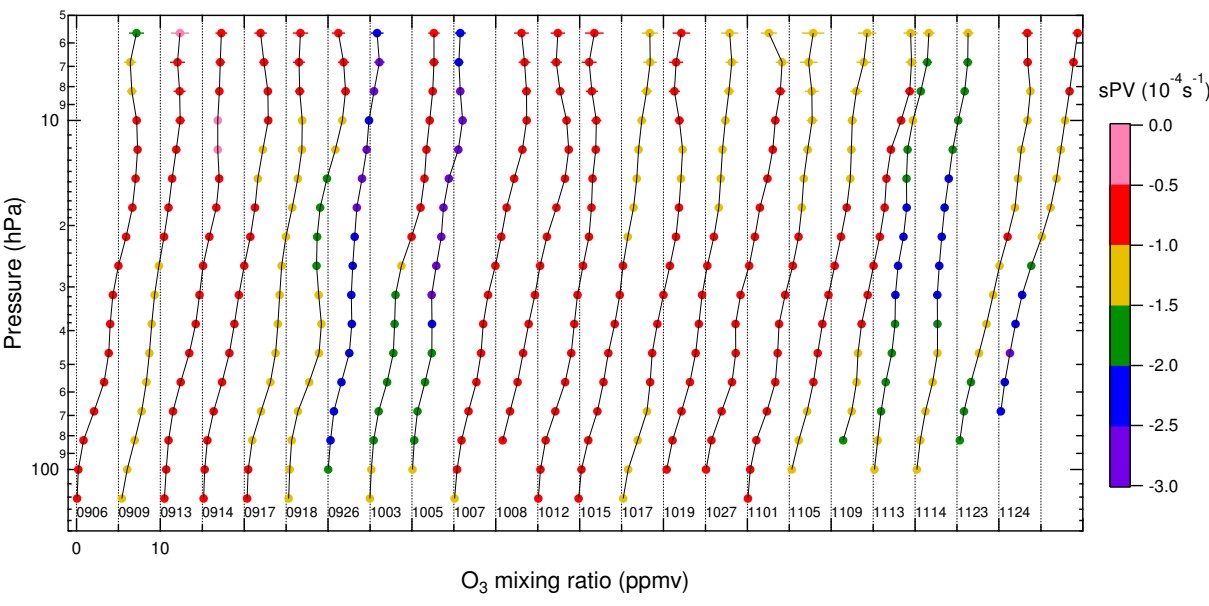

**Figure 3.** Time series of DIAL $O_3$ profiles at the OAPA site. Each profile is shifted 5 ppmv. Data are color-scaled based on sPV values. Observation dates in 2009 are shown as MMDD, e.g., 0906 is September 6, 2009.

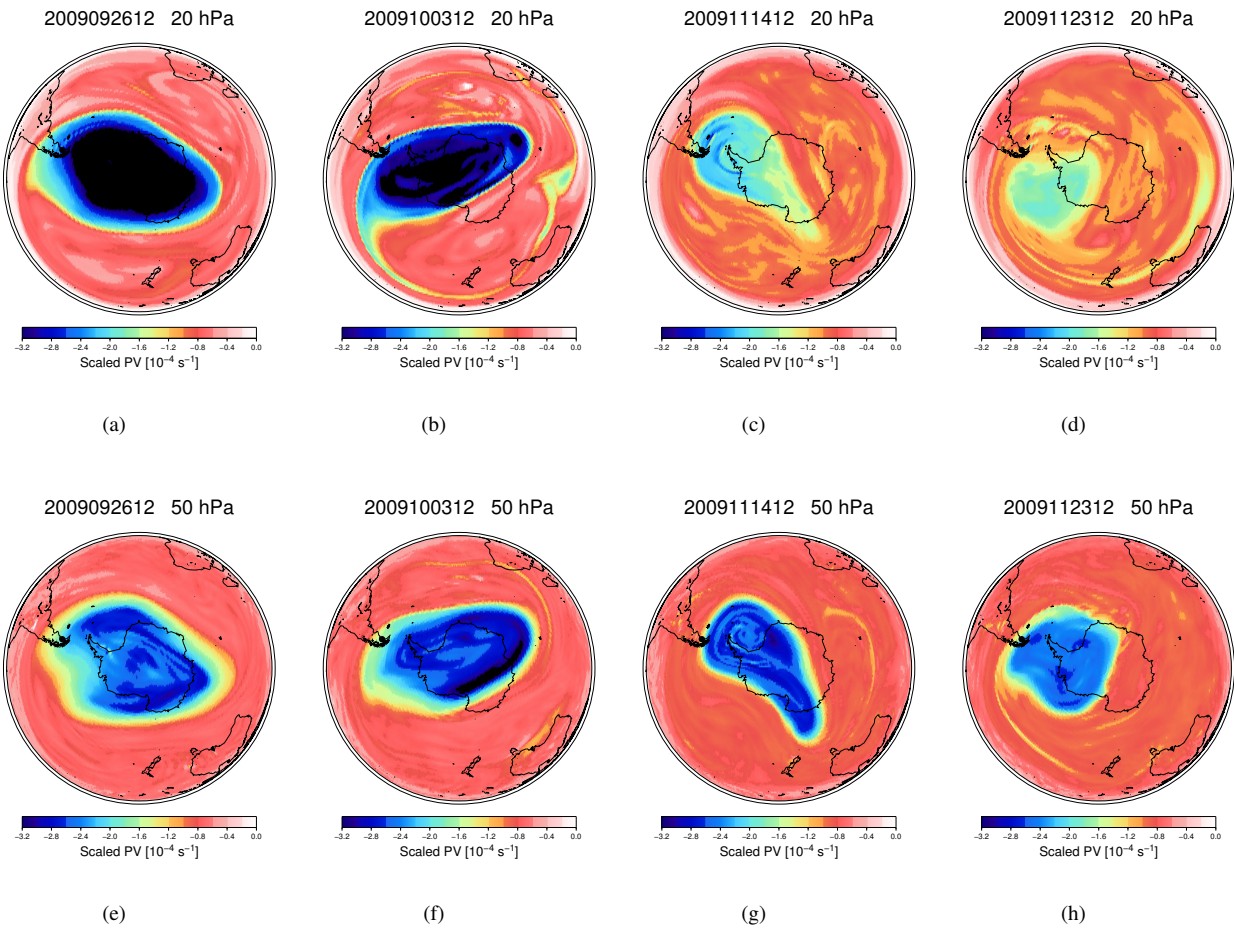

**Figure 4.** Scaled PV maps from MERRA-2 on September 26 (a, e), October 3 (b, f), November 14 (c, g), and November 23, 2009 (d, h). Top and bottom rows show pressure surfaces at 20 hPa and 50 hPa, respectively.

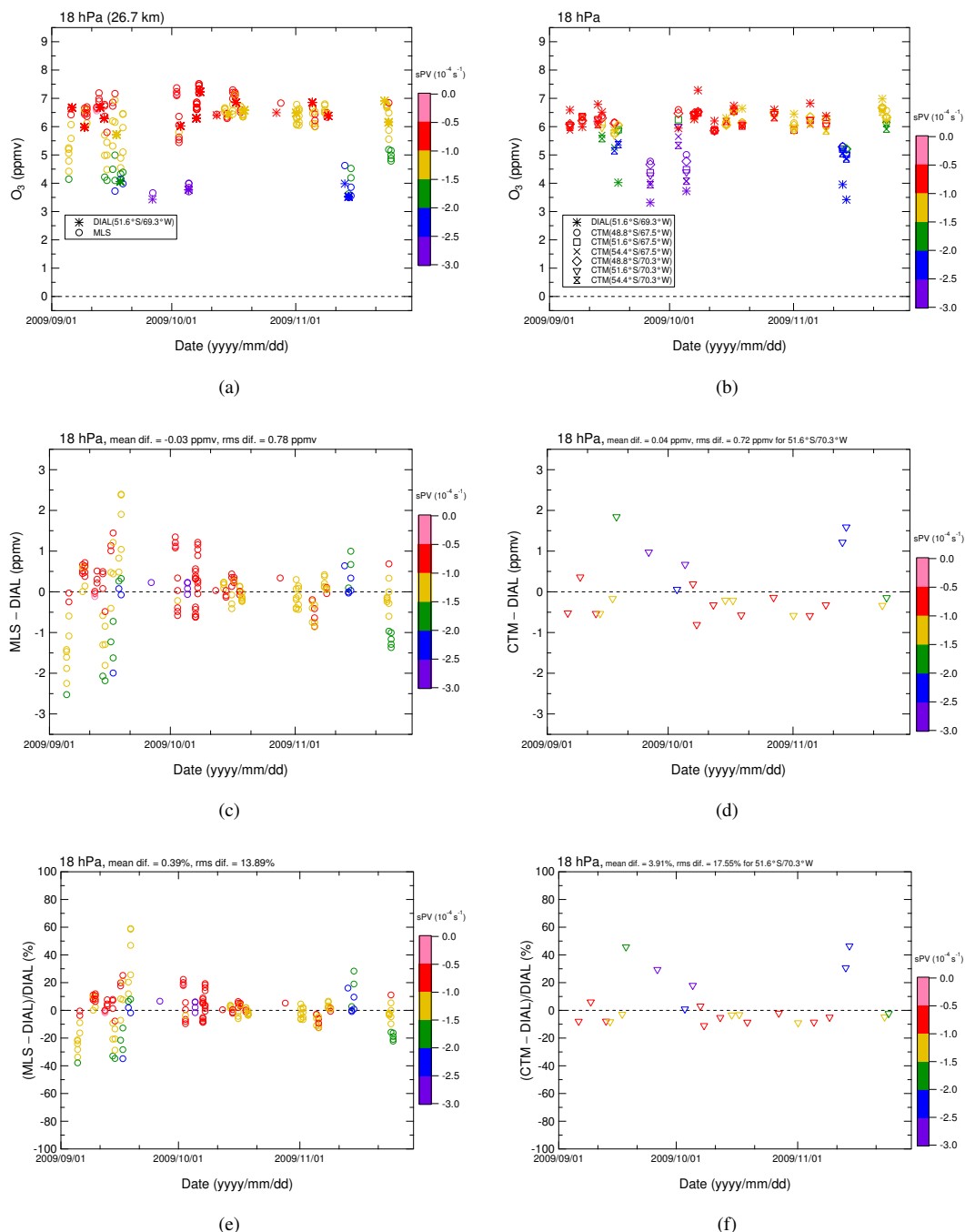

**Figure 5.** Time series of O$_3$ mixing ratios as measured by DIAL and MLS at 18 hPa (a) and absolute and relative differences between the two (c, e) from September to November 2009, over the OAPA site. (b) and (d, f) are same as (a) and (c, e), but for DIAL and MIROC-CTM. Data are color-scaled based on sPV values. For the absolute and relative differences, sPV values for MLS and MIROC-CTM are color-scaled. For MIROC-CTM, outputs from six grids are shown (see text for additional description).

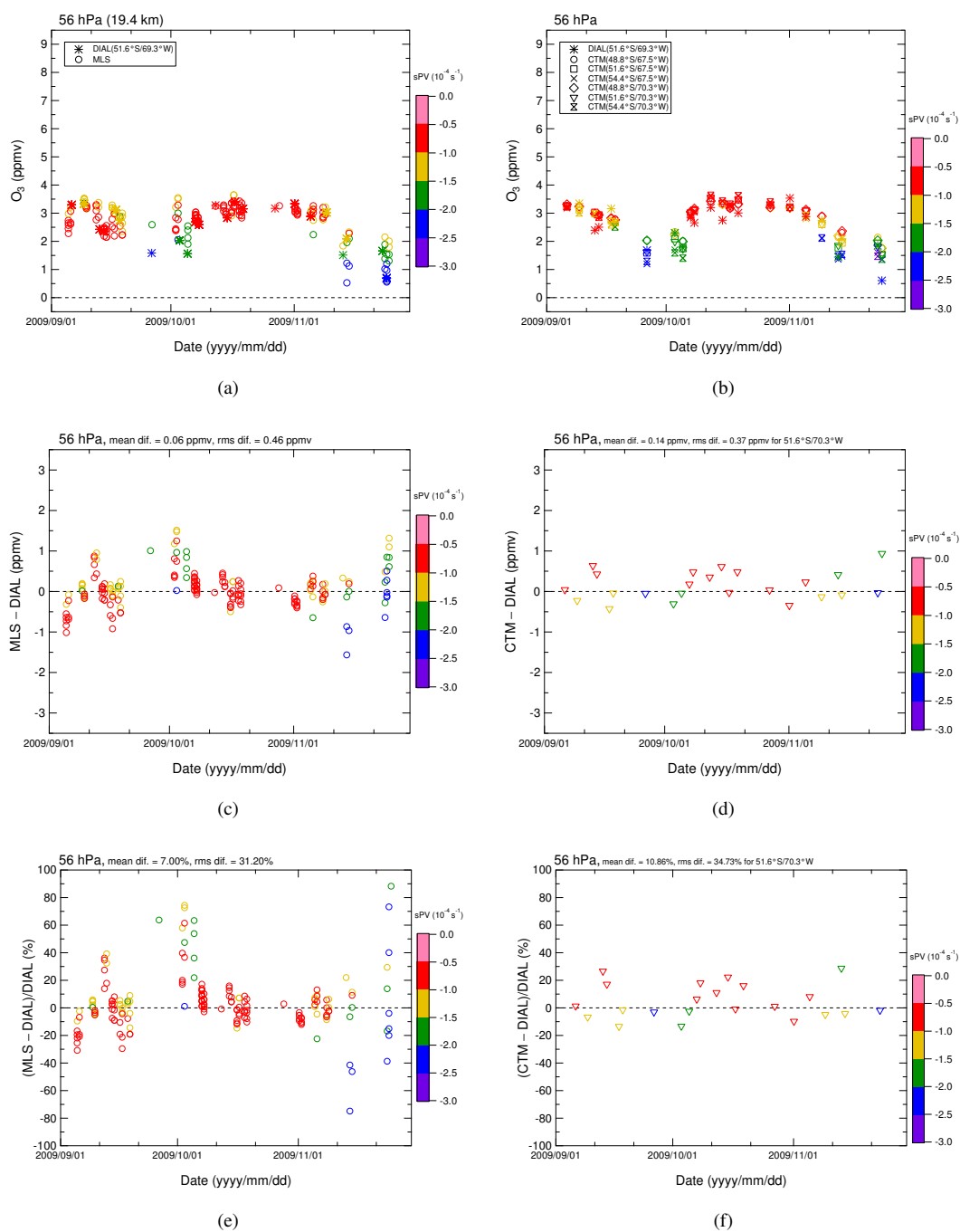

**Figure 6.** Time series of O$_3$ mixing ratios as measured by DIAL and MLS at 56 hPa (a) and absolute and relative differences between the two (c, e) from September to November 2009, over the OAPA site. (b) and (d, f) are same as (a) and (c, e), but for DIAL and MIROC-CTM. Data are color-scaled based on sPV values. For the absolute and relative differences, sPV values for MLS and MIROC-CTM are color-scaled. For MIROC-CTM, outputs from six grids are shown (see text for additional description).

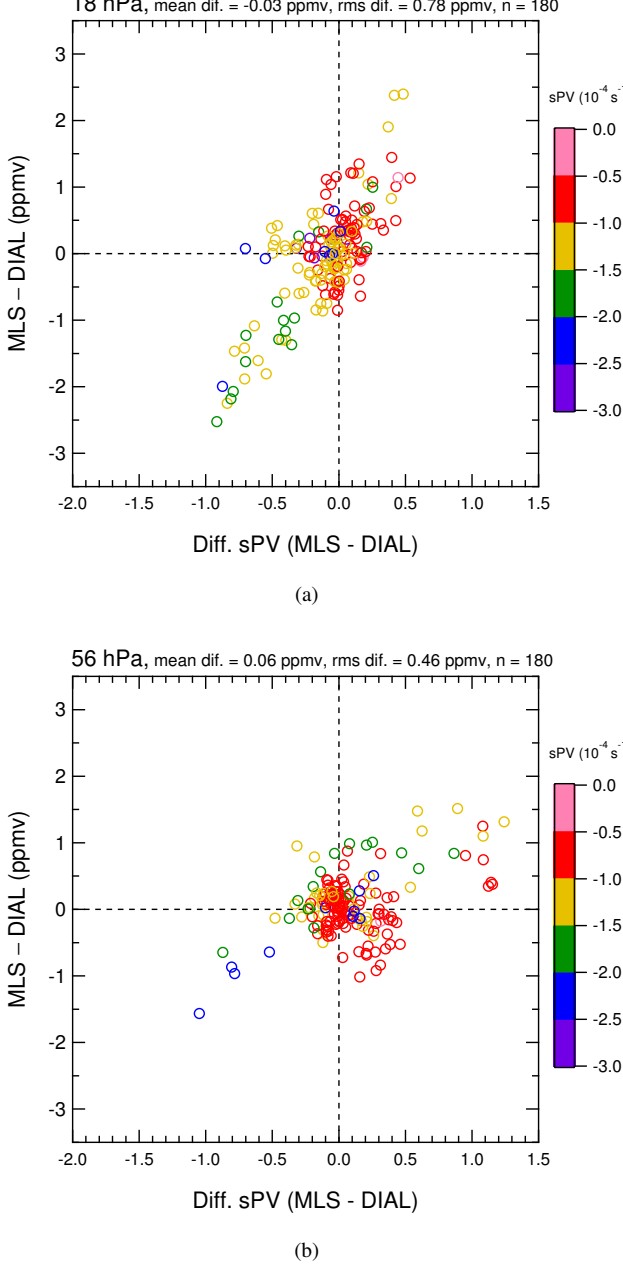

**Figure 7.** O$_3$ difference versus sPV difference for DIAL and MLS at 18 hPa (a) and 56 hPa (b). Data are color-scaled based on the sPV for the MLS measurements.

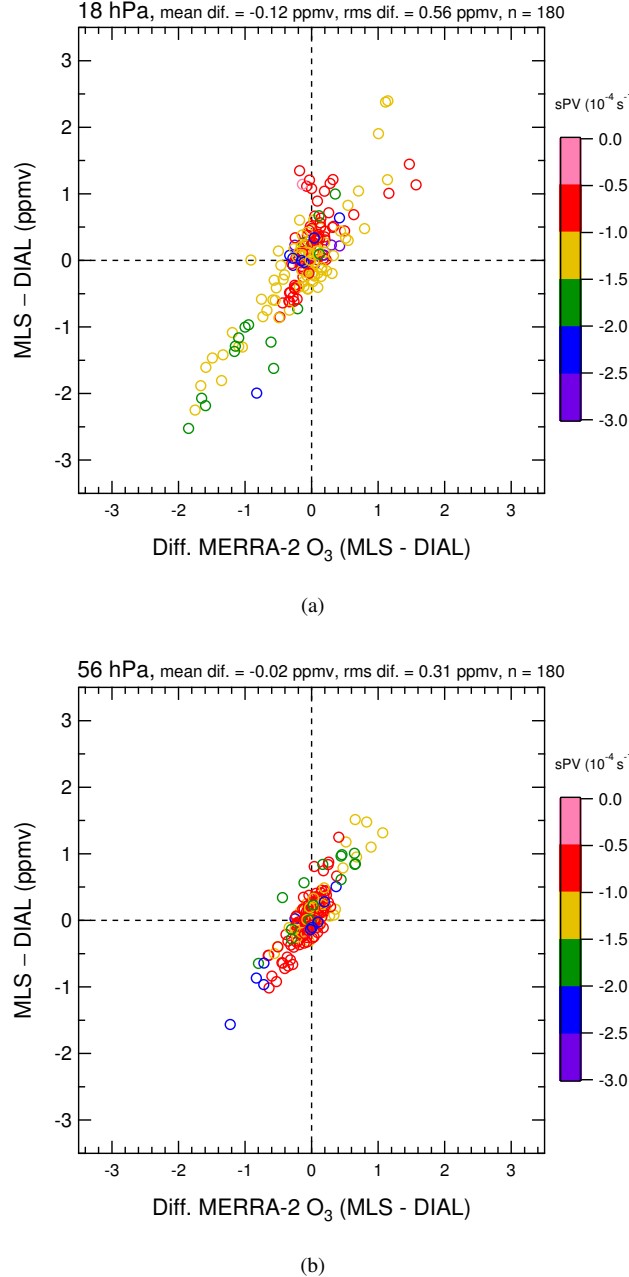

**Figure 8.** O$_3$ difference versus MERRA-2 O$_3$ difference for DIAL and MLS at 18 hPa (a) and 56 hPa (b). Data are color-scaled based on the sPV for the MLS measurements.

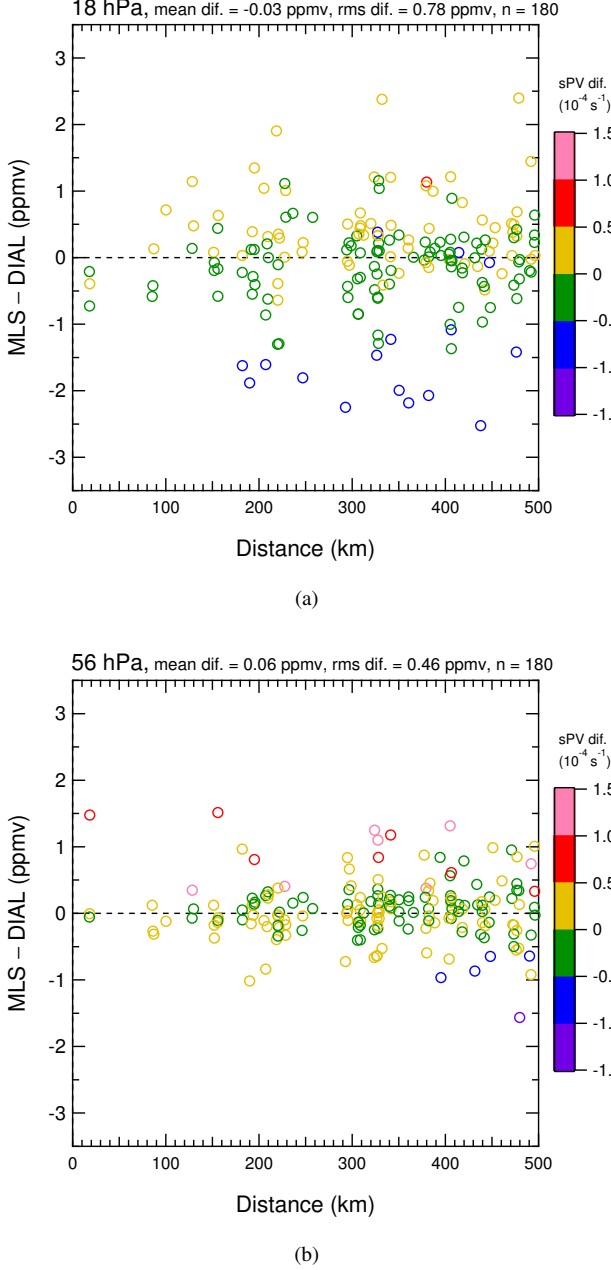

**Figure 9.** O$_3$ difference versus distance for DIAL and MLS at 18 hPa (a) and 56 hPa (b). Data are color-scaled based on sPV differences between DIAL and MLS measurements.

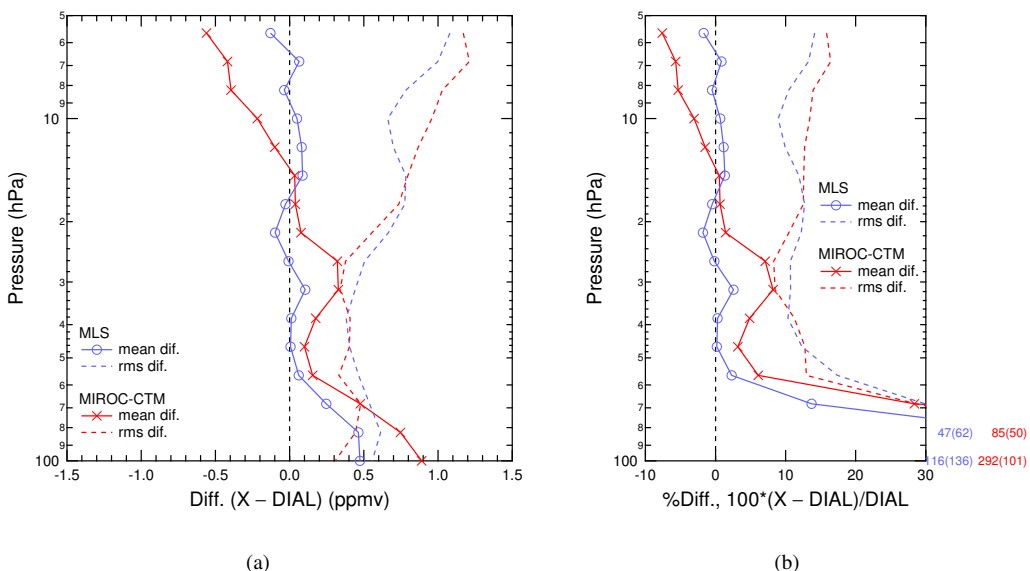

(a)                    (b)

**Figure 10.** Vertical profiles of mean and rms differences of $O_3$ values for DIAL and X (MLS or MIROC-CTM) (a) and those of relative values (b). Each value is computed from each pressure level in the time series as shown in Figures 5 and 6. The numbers outside the plot are values of the mean (rms in parentheses) difference at 83 hPa and 100 hPa.

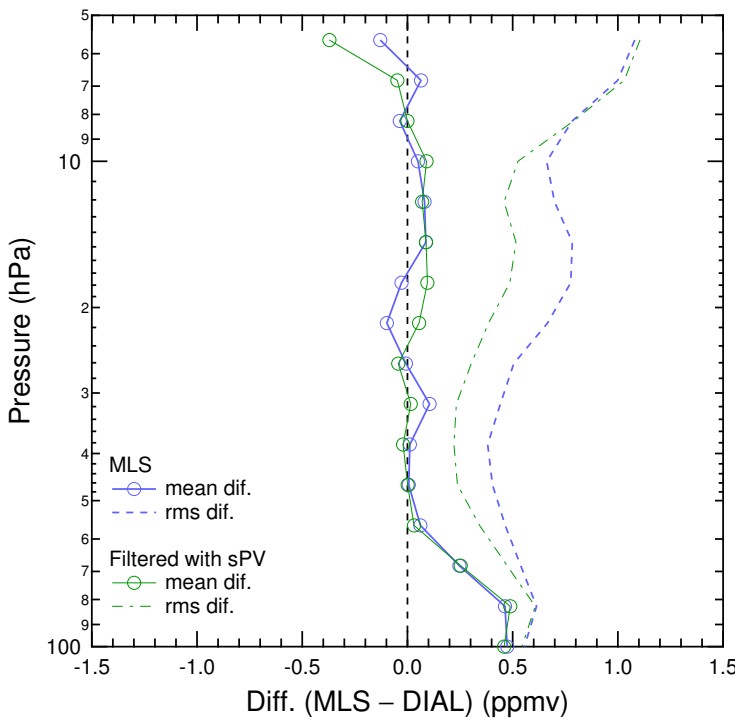

**Figure 11.** Vertical profiles of mean and rms differences with and without scaled PV criterion screening for the DIAL/MLS O$_3$ comparison.

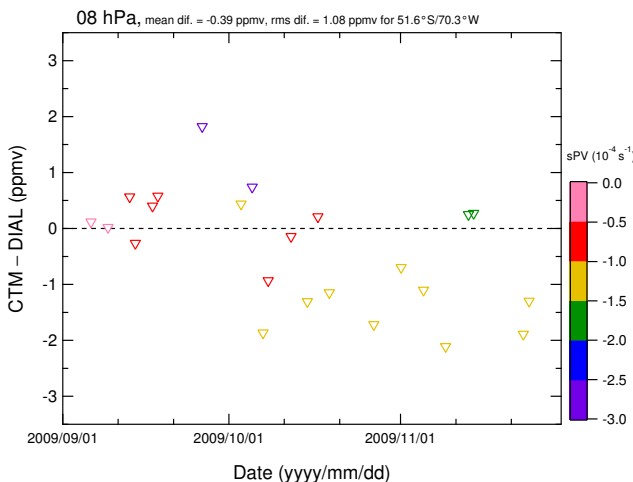

**Figure 12.** Time series of differences in $O_3$ mixing ratios as measured by DIAL and computed by MIROC-CTM at 8 hPa from September to November 2009, over the OAPA site. Data are color-scaled based on sPV values for MIROC-CTM.