# Peer review of "Comparison of ozone profiles from DIAL, MLS, and chemical transport model simulations over Río Gallegos, Argentina during the spring Antarctic vortex breakup, 2009"

_Atmospheric Measurement Techniques, 2017_

## Referee Comment (RC1) · Anonymous Referee #1 · 6 Sep 2017

General Summary:

This study evaluates the agreement between DIAL profiles, MLS, and CTM in austral spring in Rio Gallegos, Argentina. The material is appropriate for AMT and I have provided some comments below to improve the overall manuscript. Additional references should be added. A location map would improve the understanding of the manuscript in the context of the polar vortex.

Technical Comments:

[Figure]

A site lat/lon map would improve the discussion in the introduction surrounding the location of the site and vicinity to the polar vortex. Including a map with model/satellite overlay during the case studies would also improve the understanding of the horizontal scale of the variability within the latitude bands of interest.

P2L27 -It seems you have already evaluated the DIAL, this is more of an evaluation of the MLS/CTM.

P3L6 -The reference Hubert et al., 2016 is used quite extensively. As DIAL has a very robust heritage, consider referring to additional investigators in locations such as this.

P3L18-24 -This should be more concisely written as it is very qualitative. Backscatter is from the entire atmosphere, including aerosols. Consider adding in further references.

P3L28 -'horizontal spatial resolution' – what altitude are you assuming this wind field at? Later on the manuscript interprets differences due to spatial locations, this seems counter to that.

P3L30 – Total measurement uncertainty – can you describe this more? Does this involve the uncertainty from ozone absorption cross-section, Pulse-pile up, background subtraction? Are you using the retrieved MLS temperature and number density for these comparisons (i.e. ruling out metadata as a source of difference)? Is for a 3 or 5 hour measurement?

If there is large uncertainty in the DIAL measurements below a certain altitude range, consider removing them from the manuscript.

P5L14 – 'In this study step one was not necessary' - then remove this discussion, confusing for reader.

P5L16 – specify great circle lat/lon

P5L30 – Was eq.1 used or not? If so, revisit this section. If not, drop Eq. 1 entirely from manuscript.

Fig 1. – I understand the need for the MLS profiles, but are all of the CTM profiles necessary? They seem to cause more confusion and are not even compared in the difference plot. Also, the CTM profiles are significantly different above 10 hPa, this should be discussed.

P6L5 – add in plots of MLS potential temperature in Fig 1 if that is the case

P6L25 – Consider adding in a sentence for the reader to better understand PV and the relationship to the polar vortex. A map of the vortex using scaled PV would be helpful to understand why/where the boundaries were drawn on certain days.

Does MIROC-CTM provide a PV product? Is MERRA-2 meteorological variables driving the CTM? If not, couldn't differences in the modeled PV be driving large differences? MERRA-2 provides ozone as well – if this is used, why not compare it as well?

There is significant vertical motion occurring during the polar vortex breakup, is it worth looking at more vertical levels than just two to evaluate MLS/CTM? Lidar is powerful at analyzing the entire vertical profile. It would be useful to isolate a series of lidar profiles that demonstrate the variability in the polar vortex sPV regimes. Fig 5 highlights the differences that may be associated with horizontal differences, but there is no mention of how the vertical gradients may affect the overall differences between measurements.

---

## Referee Comment (RC2) · Anonymous Referee #2 · 26 Sep 2017

This article describes stratospheric ozone comparisons between the differential absorption lidar (DIAL) measurements of Rio Gallegos, Argentina, the Aura Microwave Limb Sounder (MLS) satellite observations, and the MIROC Chemistry-Transport Model (MIROC-CTM) outputs. The manuscript is well-written, and contains results from rare observations from a southern hemisphere ground-based station, making this contribution worth-publishing, after the few minor comments listed below can be addressed adequately.

Page 5, lines 8-15: What altitude variable is being used for conversion to pressure? (is

[Figure]

it the MLS-provided geopotential height?) Is geopotential height converted to geometric altitude? Please provide more details here

Figure 1, right panel: Can the combined (MLS and DIAL) uncertainty be added to the plot. This woudl show the differences int he context of their uncertainty estimates

Figures 2 and figures 3: Please add approximate geometric altitude for convenience Also, I would recommend showing differences in percent as well.

Page 7, lines 26-29: The explanation of model high bias is not convincing. Could the bias be related to inaccurate/incomplete chemistry causing in-vortex ozone loss to be underestimated? Please provide additional details supporting this statement.

On the use of meteorological fields: MIROC-CTM apparently uses ERA-based meteorological fields. However meteorological fields from GEOS-5/MERRA-2 are used for the other work described here (PV calculation, pressure/altitude conversion etc.). Would it be possible to use the same dynamical fields for improved consistency? If not, some discussion on the implications of using different met fields should be added, for example in section 3.

Figure 6b, (X-DIAL)/(X+DIAL)*200: I think plotting differences between instruments should not be done with respect to the mean of the 2 instruments. Biases between instruments are better identified when one instrument is used as the reference (typically, the instrument believed to have a best accuracy). I would recommend to modify figure 6b by taking DIAL as the reference, i.e., plot (X-DIAL)/DIAL*100 instead.

Page 9, lines 26-34: Below 70 hPa, large percent differences between observations are typically expected due to the lower ozone mixing ratio values at the bottom of the stratosphere, and occasionally also due to the proximity of the tropopause. The lidar signal saturation is a possible reason for the low bias, but the large percent differences are likely associated with the loss of sensitivity in this region of low ozone concentration

Conclusion: There is little discussion on the CTM outputs, especially the low ozone

bias inside the vortex at 18 hPa. This finding deserves some digging to my opinion, including references to published works on the subject. Finally the conclusion should emphasize the crucial importance of the DIAL station location and the dearly-needed continuation for long-term measurements there for NDACC.

---

## Author Comment (AC1) · 2 Nov 2017

General Summary:

*"This study evaluates the agreement between DIAL profiles, MLS, and CTM in austral spring in Rio Gallegos, Argentina. The material is appropriate for AMT and I have provided some comments below to improve the overall manuscript. Additional references should be added. A location map would improve the understanding of the manuscript in the context of the polar vortex."*

**Reply**: We thank the referee for the effort to carefully reading the manuscript and providing us useful comments. All of the comments are considered properly as listed below. Several references are included properly. A map showing the DIAL observation site in Río Gallegos is also included.

A list below shows differences in Figures after and before this revision.

| New | Old |
|---|---|
| Fig.1: map of Rio Gallegos | |
| Fig.2: vertical profiles (example) | Fig.1: vertical profiles (example) |
| Fig.3: time series of DIAL profiles | |
| Fig.4: sPV maps | |
| Fig.5: time series (18hPa) | Fig.2: time series (18hPa) |
| Fig.6: time series (56hPa) | Fig.3: time series (56hPa) |
| Fig.7: abs diff. vs. sPV diff. | Fig.4: abs diff. vs. sPV diff. |
| Fig.8: abs diff. vs. MERRA-2 O3 diff. | |
| Fig.9: abs diff. vs. distance | Fig.5: abs diff. vs. distance |
| Fig.10: abs/rel diff. vs. prs | Fig.6: abs/rel diff. vs. prs |
| Fig.11: abs diff. (w/filtered) vs. prs | Fig.7: abs diff. (w/filtered) vs. prs |
| Fig.12: time series of abs diff. (8hPa) | |

Technical Comments:

*1. "A site lat/lon map would improve the discussion in the introduction surrounding the location of the site and vicinity to the polar vortex. Including a map with model/satellite overlay during the case studies would also improve the understanding of the horizontal scale of the variability within the latitude bands of interest."*

**Reply:** The new Figure 1 shows location of Río Gallegos. We have added a sentence in Section 1 Introduction: "A map showing the OAPA site is shown in Figure 1." A map with model/satellite overlay for one case study on Oct. 3, 2009 is shown in Figure S2. We would like to leave this figure as it is in Supplement.

*2. "P2L27 -It seems you have already evaluated the DIAL, this is more of an evaluation of the MLS/CTM."*

**Reply:** This DIAL system in Río Gallegos is not extensively evaluated so far (Wolfram et al, 2008; Wolfram et al, 2012). The MLS ozone data have shown the long-term stability and the small bias relative to ozonesonde and ozone lidar (Hubert et al., 2016). Thus, this DIAL ozone data will not be dedicated for validating the MLS ozone data, but the comparison is a good opportunity to evaluate the inter-consistency. On the other hand, the DIAL/MIROC-CTM comparison will be dedicated for evaluating performance of the model in the southern polar vortex season where we have no such a model comparison so far.

*3. "P3L6 -The reference Hubert et al., 2016 is used quite extensively. As DIAL has a very*

*robust heritage, consider referring to additional investigators in locations such as this."*

**Reply:** We have revised these sentences: "DIAL is a laser-based active remote sensing system operated from the ground, aircraft, and ship, and has a robust heritage (e.g., Megie et al., 1977; Browell et al., 1983; Steinbrecht et al., 1989). O3 measurements from DIAL have a high vertical resolution and measurements have shown long-term stability (Nair et al., 2012; Hubert et al., 2016), owing to the stratospheric ozone lidar sites of NDACC (e.g., Leblanc and McDermid, 2000; Brinksma et al., 2002; Godin-Beekmann et al., 2003, Steinbrecht et al., 2009)."

**4.** *"P3L18-24 - This should be more concisely written as it is very qualitative. Backscatter is from the entire atmosphere, including aerosols. Consider adding in further references."*

**Reply:** We have revised a corresponding sentence: "The O3 number density profile is computed using the DIAL equation from the difference between the signal slopes originating from Rayleigh scattering of the emitted laser beams ($n_{O3}$). Since the returned signals include scattering and attenuation by atmospheric molecules, aerosols, and other atmospheric components, this complementary term could be minimized with laser wavelength chosen in the DIAL instrument. The laser wavelength chosen in the DIAL instrument minimizes the complementary term in the stratosphere to less than 10% of $n_{O3}$ measured, in the presence of low aerosol loading (Pelon et al., 1986)."

**5.** *"P3L28 -'horizontal spatial resolution' – what altitude are you assuming this wind field at? Later on the manuscript interprets differences due to spatial locations, this seems counter to that."*

**Reply:** This is referred to some typical conditions in the lower stratosphere, so that we have evaluated horizontal distance using an air-parcel trajectory analysis at 83 hPa and summarized in new Table S1. The measurement duration for each date is also summarized in new Table S1. This sentence was revised as: "Most measurements were performed for 3-5 h to obtain a good signal-to-noise ratio (see Table S1 for detailed numbers). If we assume some typical wind speed of 30 m/s in the lower stratosphere, a horizontal spatial resolution becomes 300-500 km. In actual, we have evaluated horizontal distances using air-parcel trajectory analysis at 83 hPa (Tomikawa and Sato, 2005) and the results are summarized in Table S1."

**6.** *"P3L30 – Total measurement uncertainty – can you describe this more? Does this involve the uncertainty from ozone absorption cross-section, Pulse-pile up, background subtraction? Are you using the retrieved MLS temperature and number density for these comparisons (i.e. ruling out metadata as a source of difference)? Is for a 3 or 5 hour measurement?"*

**Reply:** We have added sentences in Section 3 Method for comparisons between DIAL and MLS/CTM: "For the total measurement uncertainty (Wolfram et al., 2008), we evaluated the effect of ozone absorption cross section, which is temperature dependent, and found the error is not larger than 2%. The other source is from correction of aerosol contamination. The methodology uses a Fernald inversion algorithm to evaluate the aerosol backscatter signal at 355 nm and extrapolated to 308 nm. In order to increase the signal to noise ratio, the signal registered is averaged over the full acquisition time of the measurement. The acquisition time is typically three to four hours, according to weather conditions. Before processing the signal using the DIAL equation, we make two corrections: 1) subtraction of the background signal using a linear regression within the range of altitudes where the lidar signal is considered negligible, typically between 80 and 150 km; 2) dead time correction of the detector, in order to correct the saturation of the photocounting signals (pile-up effect) in the lower altitude ranges (Godin et al., 1999)."

We do not use MLS temperature and pressure for converting O3 number density to mixing ratio. We used temperature, pressure, and geopotential height of the NCEP reanalysis data (Kalnay et al., 1996) for conversion. Thus, this is another source of uncertainty for the pressure/mixing ratio coordinate comparison. We have added a sentence in Section 3 Method for comparisons between DIAL and MLS/CTM: "For converting the original DIAL geometric altitude and O3 number density to pressure and O3 mixing ratio, the NCEP reanalysis data

(Kalnay et al., 1996) are used. These data are registered in the NDACC database. Possible deviations could be expected if we use other meteorological data for the conversion process in DIAL. However, in this study, we used the DIAL data that registered in the NDACC database."

  The actual measurement duration is listed in new Table S1, ranging from 02:24 to 05:45.

***7.*** *"If there is large uncertainty in the DIAL measurements below a certain altitude range, consider removing them from the manuscript."*
**Reply:** As in new Figure 3, the total error is not so large, but the O3 mixing ratio itself becomes small values in the lowermost stratosphere, providing large relative difference. Thus, we will leave the results of 83 hPa and 100 hPa levels.

***8.*** *"P5L14 – 'In this study step one was not necessary' - then remove this discussion, confusing for reader."*
**Reply:** We have deleted the corresponding sentences and simply mention as below: "For comparison between DIAL and MLS, the DIAL profile is convolved using the following equation (Livesey et al., 2017): ..."

***9.*** *"P5L16 – specify great circle lat/lon"*
**Reply:** It becomes between 47.1S and 56.1S for 69.3W. These values were added in the sentence.

***10.*** *"P5L30 – Was eq.1 used or not? If so, revisit this section. If not, drop Eq. 1 entirely from manuscript."*
**Reply:** Eq. 1 was used. We have revised sentences: "Figure 2a shows vertical profiles of O3 measured with DIAL compared with those of MLS on the same day (November 14, 2009) as an example. The plus-crosses and dotted-line show the converted DIAL profile using Equation (1) and the original high vertical resolution DIAL profile, respectively."

***11.*** *"Fig 1. – I understand the need for the MLS profiles, but are all of the CTM profiles necessary? They seem to cause more confusion and are not even compared in the difference plot. Also, the CTM profiles are significantly different above 10 hPa, this should be discussed."*
**Reply:** Figure 1 has revised as new Figure 2. CTM profile nearest the OAPA site is shown. We have also revised a corresponding sentence: "We have extracted data from six locations between 48.8S and 54.4S in latitude at 67.5W and 70.3W in longitude, but the nearest grid data was plotted in Figure 2a (see Figures 5 and 6 for the variability in six model grids)."

  Above 10 hPa, the deviation of the CTM profile found on November 23, 2009 was actually discussed in Section 4.4 Comparison at other levels. So that, we have added a sentence in the last of Section 4.1 Example of vertical profile comparison: "This is discussed in Section 4.4."

***12.*** *"P6L5 – add in plots of MLS potential temperature in Fig 1 if that is the case"*
**Reply:** Figure 1 has revised as new Figure 2. The potential temperatures (PT) for MLS computed from the MERRA-2 data are shown as text, instead of plots, for both new Figures 2a and 2b. Corresponding to PT levels of Wolfram et al. (2012), we show 475, 550, 650, and 850 K levels. We have also added a sentence in Section 4.1 Example of vertical profile comparison: "Several PT levels corresponding to pressure are also shown as text in Figure 2a."

***13.*** *"P6L25 – Consider adding in a sentence for the reader to better understand PV and the relationship to the polar vortex. A map of the vortex using scaled PV would be helpful to understand why/where the boundaries were drawn on certain days."*
**Reply:** We have added a sentence: "The degree of PV values at each measurement or model grid is a robust indicator of the location relative to the polar vortex." sPV maps for selected days are now shown in new Figure 4. Sentences are added in Section 4.2 Time series comparison: "Figure 4 shows sPV maps from MERRA-2 for selected days on September 26, October 3, November 14, and November 23, 2009. At 20 hPa, the polar

vortex significantly diminishes on November 23 compared to that on September 26. Whereas at 50 hPa, the polar vortex still exists on November 23 with smaller area than that on September 26."

***14.*** *"Does MIROC-CTM provide a PV product? Is MERRA-2 meteorological variables driving the CTM? If not, couldn't differences in the modeled PV be driving large differences? MERRA-2 provides ozone as well – if this is used, why not compare it as well?"*

**Reply:** We can compute the PV value from the output of MIROC-CTM, but we used PV values from MERRA-2 for location and time of all DIAL, MLS, and MIROC-CTM in this study to unify the data source. We have performed a model run with MERRA-2 but not achieved a detailed comparison yet (Akiyoshi et al., a presentation in Meteorological Society of Japan, 2017). Since a possible deviation could be expected if we use MERRA-2 for the nudging process in MIROC-CTM, some discussion has added in Section 3 Method for comparisons between DIAL and MLS/CTM: "... Another possible deviations could also be expected if we use other meteorological data for the nudging process in MIROC-CTM. The different reanalysis data may cause different vertical and horizontal motions of air in the model, providing different tracer correlations, hence ozone field. However, in this study, we analyze owing to the model of Akiyoshi et al. (2016) to examine the performance." (See also in Point 6 of Referee#2.)

We have analyzed the O3 value from MERRA-2, and added new Figure 8 and some discussion in Section 4.3 Dependency in distance and sPV difference: "Since the MERRA-2 data set also provide the O3 value (Wargan et al., 2017), we examined those data instead of the sPV value. Figure 8 shows the O3 difference versus MERRA-2 O3 difference between DIAL and MLS (MLS - DIAL). The mean difference is computed from the horizontal axis, resulting in -0.12 ppmv at 18 hPa and -0.02 ppmv at 56 hPa. The measured O3 difference is well reproduced by the MERRA-2 O3 that assimilates Aura MLS as well. At 56 hPa, a compact correlation is found between the two differences with a slope of one-by-one. A similar positive correlation is also found at 18 hPa."

***15.*** *"There is significant vertical motion occurring during the polar vortex breakup, is it worth looking at more vertical levels than just two to evaluate MLS/CTM? Lidar is powerful at analyzing the entire vertical profile. It would be useful to isolate a series of lidar profiles that demonstrate the variability in the polar vortex sPV regimes. Fig 5 highlights the differences that may be associated with horizontal differences, but there is no mention of how the vertical gradients may affect the overall differences between measurements."*

**Reply:** We have extended discussion on the low bias in MIRCO-CTM above 10 hPa (relate to Point 11). To examine the effect of the vortex breakup on the O3 difference, we have added the O3 difference time series at 8 hPa in new Figure 12. A new Figure 3 shows a series of 23 DIAL profiles. Data are color-scaled based on sPV values. With these figures, we have revised sentences in Section 4.4 Comparison at other levels: summary: "To examine the low bias in MIROC-CTM, the time-series in O3 difference between DIAL and MIROC-CTM at 8 hPa is shown in Figure 12. Larger negative deviations in MIROC-CTM are found in October and November, especially for data with sPV values between -1.0 and -1.5 x $10^{-4}$ $s^{-1}$. Similar results are also found from 6 hPa and 7 hPa levels. The peak altitude of ozone in MIROC-CTM is lower than that of DIAL, as shown in Figure 2. Both the vertical and horizontal motions of air in the model are responsible for this different feature, but the cause is not known. As was shown in Figure 3, the vertical gradient of O3 from DIAL above 15-20 hPa shows rather week inside the polar vortex, but occasionally strong outside or edge of the polar vortex. Thus, the vertical gradient of O3 may affect the result for such occasions with the steeper gradient."

Also, we have added a sentence for the new Figure 3 in Section 4.2 Time series comparison: "Figure 3 shows all the 23 profiles of O3 
[revised manuscript text omitted]

---

## Author Comment (AC2) · 2 Nov 2017

*1. "This article describes stratospheric ozone comparisons between the differential absorption lidar (DIAL) measurements of Rio Gallegos, Argentina, the Aura Microwave Limb Sounder (MLS) satellite observations, and the MIROC Chemistry-Transport Model (MIROC-CTM) outputs. The manuscript is well-written, and contains results from rare observations from a southern hemisphere ground-based station, making this contribution worth-publishing, after the few minor comments listed below can be addressed adequately."*
**Reply**: We thank the referee for the effort to carefully reading the manuscript and providing us useful comments. All of the comments are considered properly as listed below.

A list below shows differences in Figures after and before this revision.

| New | Old |
|---|---|
| Fig.1: map of Rio Gallegos | |
| Fig.2: vertical profiles (example) | Fig.1: vertical profiles (example) |
| Fig.3: time series of DIAL profiles | |
| Fig.4: sPV maps | |
| Fig.5: time series (18hPa) | Fig.2: time series (18hPa) |
| Fig.6: time series (56hPa) | Fig.3: time series (56hPa) |
| Fig.7: abs diff. vs. sPV diff. | Fig.4: abs diff. vs. sPV diff. |
| Fig.8: abs diff. vs. MERRA-2 O3 diff. | |
| Fig.9: abs diff. vs. distance | Fig.5: abs diff. vs. distance |
| Fig.10: abs/rel diff. vs. prs | Fig.6: abs/rel diff. vs. prs |
| Fig.11: abs diff. (w/filtered) vs. prs | Fig.7: abs diff. (w/filtered) vs. prs |
| Fig.12: time series of abs diff. (8hPa) | |

*2. "Page 5, lines 8-15: What altitude variable is being used for conversion to pressure? (is it the MLS-provided geopotential height?) Is geopotential height converted to geometric altitude? Please provide more details here"*
**Reply**: For conversion of the DIAL altitude/O3 number density into pressure/O3 mixing ratio, we used the NCEP reanalysis data, and the data are registered in NDACC. The geopotential height of the NCEP data is converted to geometric altitude. We have added a sentence in Section 3 Method for comparisons between DIAL and MLS/CTM: "For converting the original DIAL geometric altitude and O3 number density to pressure and O3 mixing ratio, the NCEP reanalysis data (Kalnay et al., 1996) are used. These data are registered in the NDACC database."

*3. "Figure 1, right panel: Can the combined (MLS and DIAL) uncertainty be added to the plot. This would show the differences in the context of their uncertainty estimates"*
**Reply**: We missed to mention the bars in MLS O3 profiles in the submitted version. This is precision reported for individual profiles. The combined uncertainty is added in new Figure 2 (right panel). The total uncertaintiy for DIAL is also added in new Figure 2 (left panel). We have added sentences in Section 4.1 Example of vertical profile comparison: "The bar in MLS O3 profiles shows the precision reported for individual profiles. The bar in DIAL O3 profile shows the total uncertainty. The combined uncertainty (root sum square) is shown in the right panel."

*4. "Figures 2 and figures 3: Please add approximate geometric altitude for convenience Also, I would recommend showing differences in percent as well."*
**Reply**: Approximate geometric altitude of DIAL are now added in new Figure 5 and Figure 6, and relative differences as 100*(X - DIAL) / DIAL are also added as another panels in new Figure 5 and Figure 6. We have added sentences in Section 4.2 Time series comparison: "For reference, Figures 5e and 5f show the relative differences for DIAL/MLS and DIAL/MIROC-CTM

comparisons, respectively." and "Figures 6e and 6f show the relative differences for DIAL/MLS and DIAL/MIROC-CTM comparisons, respectively."

**5.** *"Page 7, lines 26-29: The explanation of model high bias is not convincing. Could the bias be related to inaccurate/incomplete chemistry causing in-vortex ozone loss to be underestimated? Please provide additional details supporting this statement."*
**Reply**: For this high bias in the MIROC-CTM, one possible explanation was as follow: A higher $N_2O$ value at 18 hPa than that of MLS was seen. A higher $N_2O$ value corresponds a smaller value of Cly (ClOx), providing a higher $O_3$ value owing to a weaker $O_3$ destruction. We have added some sentences in Section 4.2 Time series comparison: "Another possible explanation could be due to a weaker vertical motion of air in MIROC-CTM. Although not shown, a vertical profile of nitrous oxide, $N_2O$, from MIROC-CTM on November 14, 2009 is different from that from MLS. A tight correlation between $N_2O$ and Cly is found in the stratosphere (e.g., Schauffler et al., 2003), and used to infer the Cly value from a measured $N_2O$ value (e.g., Wetzel et al., 2010; Strahan et al., 2014). At 18 hPa, the MIROC-CTM $N_2O$ value is higher than that of MLS, resulting in a smaller value of Cly in MIROC-CTM. Thus, a smaller active chlorine (ClOx) induces a higher $O_3$ amount in MIROC-CTM."

**6.** *"On the use of meteorological fields: MIROC-CTM apparently uses ERA-based meteorological fields. However meteorological fields from GEOS-5/MERRA-2 are used for the other work described here (PV calculation, pressure/altitude conversion etc.). Would it be possible to use the same dynamical fields for improved consistency? If not, some discussion on the implications of using different met fields should be added, for example in section 3."*
**Reply**: We did not use MERRA-2 for the conversion of the DIAL altitude/$O_3$ number density to pressure/$O_3$ mixing ratio. The difference between NCEP and MERRA-2 will affect the DIAL pressure/$O_3$ mixing ratio. This will be done in a future work when we register such data to NDACC.
  Regarding a nudging other meteorological data, we performed CTM runs using MERRA-2 and NCEP reanalysis data. However, we have not achieved a detailed comparison among those meteorological data (Akiyoshi et al., a presentation in Meteorological Society of Japan, 2017). This will also be done in a future work. We have added some sentences in Section 3 Method for comparisons between DIAL and MLS/CTM: "Possible deviations could be expected if we use other meteorological data for the conversion process in DIAL. However, in this study, we used the DIAL data that registered in the NDACC database. Another possible deviations could also be expected if we use other meteorological data for the nudging process in MIROC-CTM. The different reanalysis data may cause different vertical and horizontal motions of air in the model, providing different tracer correlations, hence ozone field. However, in this study, we analyze owing to the model of Akiyoshi et al. (2016) to examine the performance."

**7.** *"Figure 6b, (X-DIAL)/(X+DIAL)*200: I think plotting differences between instruments should not be done with respect to the mean of the 2 instruments. Biases between instruments are better identified when one instrument is used as the reference (typically, the instrument believed to have a best accuracy). I would recommend to modify figure 6b by taking DIAL as the reference, i.e., plot (X-DIAL)/DIAL*100 instead."*
**Reply**: The right panel of Figure 6 has been revised in new Figure 10. According to this revision, the numbers shown in Section 4.4 Comparison at other levels: summary and in Section 5 Conclusions have revised to 116% for DIAL/MLS and 292% for DIAL/MIROC-CTM.

**8.** *"Page 9, lines 26-34: Below 70 hPa, large percent differences between observations are typically expected due to the lower ozone mixing ratio values at the bottom of the stratosphere, and occasionally also due to the proximity of the tropopause. The lidar signal saturation is a possible reason for the low bias, but the large percent differences are likely associated with the loss of sensitivity in this region of low ozone concentration"*
**Reply**: Thank you for pointing out this. We have added a sentence: "Since the $O_3$ mixing ratio from DIAL is very small below about 70 hPa, the sensitivity might be degraded along with the saturation effect."

*9.* *"Conclusion: There is little discussion on the CTM outputs, especially the low ozone bias inside the vortex at 18 hPa. This finding deserves some digging to my opinion, including references to published works on the subject. Finally the conclusion should emphasize the crucial importance of the DIAL station location and the dearly-needed continuation for long-term measurements there for NDACC."*

**Reply**: From a view of the mean difference between DIAL and CTM, there is a low bias in CTM above 18 hPa, as shown in Figure 6 (new Figure 10). However, looking at data inside the vortex, there are high biases in CTM, as shown in Figure 2d (new Figure 5d). As I mentioned above, the high biases may be associated with a weaker vertical motion of air in CTM. This partly cancelled the underestimate of O3 value, providing a mean difference of nearly zero (0.04 ppmv in Figure 2d, i.e., new Figure 5d). We have added a sentence in Section 5 Conclusions: "An insufficient model vertical motion may also be partly responsible for the O3 differences, especially inside the polar vortex."

We have also added a sentence to emphasize the continuation of DIAL observations: "Because of very sparse observations from S.H. ground-based stations, continuation for long-term measurements there for NDACC is highly recommended."

Referencers:

Akiyoshi, H., Nakamura, T., Miyasaka, T., Shiotani, M., and Suzuki, M.: A nudged chemistry-climate model simulation of chemical con- stituent distribution at northern high-latitude stratosphere observed by SMILES and MLS during the 2009/2010 stratospheric sudden warming, J. Geophy. Res., 121, 1361–1380, https://doi.org/10.1002/2015JD023334, http://dx.doi.org/10.1002/2015JD023334, 2016.

Kalnay, E., Kanamitsu, M., Kistler, R., Collins, W., Deaven, D., Gandin, L., Iredell, M., Saha, S., White, G., Woollen, J., Zhu, Y., Leet- maa, A., Reynolds, R., Chelliah, M., Ebisuzaki, W., Higgins, W., Janowiak, J., Mo, K. C., Ropelewski, C., Wang, J., Jenne, R., and Joseph, D.: The NCEP/NCAR 40-Year Reanalysis Project, Bull. American Meteorol. Soc., 77, 437–471, https://doi.org/10.1175/1520- 0477(1996)077<0437:TNYRP>2.0.CO;2, https://doi.org/10.1175/1520-0477(1996)077<0437:TNYRP>2.0.CO;2, 1996.

Schauffler, S. M., Atlas, E. L., Donnelly, S. G., Andrews, A., Montzka, S. A., Elkins, J. W., Hurst, D. F., Romashkin, P. A., Dutton, G. S., and Stroud, V.: Chlorine budget and partitioning during the Stratospheric Aerosol and Gas Experiment (SAGE) III Ozone Loss and Validation Experiment (SOLVE), J. Geophys. Res., 108, ACH 7–1–ACH 7–18, https://doi.org/10.1029/2001JD002040, http://dx.doi.org/10.1029/ 2001JD002040, 2003.

Strahan, S. E., Douglass, A. R., Newman, P. A., and Steenrod, S. D.: Inorganic chlorine variability in the Antarctic vortex and implications for ozone recovery, J. Geophys. Res., 119, 14 098–14 109, https://doi.org/10.1002/2014JD022295, http://dx.doi.org/10.1002/2014JD022295,2014.

Wetzel, G., Oelhaf, H., Kirner, O., Ruhnke, R., Friedl-Vallon, F., Kleinert, A., Maucher, G., Fischer, H., Birk, M., Wagner, G., and Engel, A.: First remote sensing measurements of ClOOCl along with ClO and ClONO$_2$ in activated and deactivated Arctic vortex conditions using new ClOOCl IR absorption cross sections, Atmos. Chem. Phys., 10, 931–945, https://doi.org/10.5194/acp-10-931-2010, 2010.

---

## Author Response (AR2)

Dear Editor,

Thank you very much for your suggestions of technical corrections as listed below.
I have revised all the points as suggested by the Associate Editor.

Also, I will receive the English Editing by ACP/AMT production before the final form.

Associate Editor Decision: Publish subject to technical corrections (05 Nov 2017) by Wolfgang Steinbrecht
Comments to the Author:
You have addressed the suggestions of both reviewers. I think the manuscript is acceptable for publication.
However, before that I do suggest the following small edits (page and line numbers refer to the change-tracked version of the manuscript):
Page 1, line 13: replace slight by small
Page 1, line 14: replace unique by important
Page 2, line 2: replace since by due to
Page 2, lines 14/15: delete "caution is needed for" and replace "to" by "are needed to"
Page 2, line 22: replace which by when it
Page 3, line 12/13: delete "generated by two emittter lasers" (It could also be one laser, e.g. a dye laser)
Page 3, line 28: replace "energy" by "intensity" (return signal are not energy, they are energy/time = intensity)
Page 6, line 22 (and everywhere else): replace color-scaled by color-coded
Page 7, line 27: replace "The difference ... the sPV" by " Ozone changes are related to the sPV"
Page 10, line 7: replace "between than" by "than with"

These were just a few examples for editing. ACP will probably copy-edit the text. This should improve clarity / english further.

Yours sincerely,

Takafumi Sugita, Dr.
National Institute for Environmental Studies
Center for Global Environmental Research
16-2 Onogawa, Tsukuba, Ibaraki 305-8506 JAPAN
Tel: +81-298-50-2460
mailto:tsugita@nies.go.jp
http://www.cger.nies.go.jp/en/